# Cover Learning for Large-Scale Topology Representation

**Luis Scoccola** [1]  **Uzu Lim** [2]  **Heather A. Harrington** [3]

## Abstract

Classical unsupervised learning methods like clustering and linear dimensionality reduction parametrize large-scale geometry when it is discrete or linear, while more modern methods from manifold learning find low dimensional representation or infer local geometry by constructing a graph on the input data. More recently, topological data analysis popularized the use of simplicial complexes to represent data topology with two main methodologies: topological inference with geometric complexes and large-scale topology visualization with Mapper graphs – central to these is the nerve construction from topology, which builds a simplicial complex given a cover of a space by subsets. While successful, these have limitations: geometric complexes scale poorly with data size, and Mapper graphs can be hard to tune and only contain low dimensional information. In this paper, we propose to study the problem of learning covers in its own right, and from the perspective of optimization. We describe a method for learning topologically-faithful covers of geometric datasets, and show that the simplicial complexes thus obtained can outperform standard topological inference approaches in terms of size, and Mapper-type algorithms in terms of representation of large-scale topology.

[1]Centre de Recherches Mathématiques et Institut des sciences mathématiques, Laboratoire de combinatoire et d'informatique mathématique de l'Université du Québec à Montréal, Université de Sherbrooke, Canada. [2]Queen Mary University of London, United Kingdom. [3]Max Planck Institute of Molecular Cell Biology and Genetics, Dresden, Germany; Centre for Systems Biology, Dresden, Germany; Faculty of Mathematics, Technische Universität Dresden, Germany; Mathematical Institute, University of Oxford, United Kingdom. Correspondence to: Luis Scoccola <luis.scoccola@gmail.com>.

*Proceedings of the 42nd International Conference on Machine Learning*, Vancouver, Canada. PMLR 267, 2025. Copyright 2025 by the author(s).

## 1. Introduction

### 1.1. Context

We are concerned with the problem of learning a representation of the large-scale structure of geometric datasets, such as point clouds. Classical instances of this problem are the clustering problem, linear dimensionality reduction, and manifold learning. More recently, building on tools from algebraic topology and computational geometry, topological data analysis (TDA) was born with the goal of learning and quantifying the topology of data beyond the discrete and linear cases. What distinguishes TDA from previous methodologies, such as classical manifold learning, is its explicit reliance on (higher-dimensional) simplicial complexes, with the stated goal often being that of producing a simplicial complex which recovers the topology of the underlying space from which the data was sampled.

Simplicial complexes (Appendix A.1) are a specific type of hypergraph, and were originally used in topology to model topological spaces combinatorially; see Figure 1 for an example. Recent applications include neuroscience (Gardner et al., 2022), biology (Benjamin et al., 2024), physics (Sale et al., 2023), and machine learning (Maggs et al., 2024).

The two main methodologies in TDA are topological inference with geometric complexes, and large-scale topology visualization with Mapper graphs; we elaborate on the limitations of these in Section 2 and Appendix A.4. In this paper, we propose *cover learning* as a unified unsupervised learning procedure which can efficiently represent the large-topology of geometric data, and simultaneously address the limitations of these two methodologies.

We now give more context and motivation for cover learning, and then describe our contributions.

### 1.2. Learning simplicial complexes

A *simplicial complex on* a set $X$ consists of a simplicial complex $K$ together with a function $K_0 \to \mathsf{Parts}(X)$ from the vertices of $K$ to the set of subsets of $X$, the idea being that vertices represent groups of data points, and higher-dimensional simplices represent relationships between these. *Simplicial complex learning* is the process of constructing a simplicial complex on an input dataset, and it encom-

passes standard constructions such as graphs in classical unsupervised learning (e.g., neighborhood graphs in Laplacian eigenmaps (Belkin & Niyogi, 2003), t-SNE (Van der Maaten & Hinton, 2008), and DBSCAN (Ester et al., 1996)), simplicial complexes in topological inference (e.g., geometric complexes (Edelsbrunner & Harer, 2022) and Mapper (Singh et al., 2007)), and meshes is surface reconstruction (Dey, 2006).

## 1.3. Learning covers

A *cover* of a set $X$ is a family $\{U_i\}_{i \in I}$ of subsets of $X$, which cover it in the sense that $X = \bigcup_{i \in I} U_i$. *Cover learning* is the process of constructing a cover of an input dataset.

The *nerve* construction from topology (Definition A.1) takes as input a cover $\mathcal{U}$ of a set $X$ and outputs a simplicial complex over $X$, denoted $\mathsf{Ner}(\mathcal{U})$. In words, the nerve of a cover has as vertices the sets of the cover, and as higher-dimensional simplices the non-empty intersections between these sets; see Figure 1 for examples. Thus, the nerve construction allows us to reduce simplicial complex learning to cover learning, and suggests the following goal:

---

> *Goal* 1 (Informal). Given a geometric dataset, cover it by local patches, in such a way that the nerve of the cover is a faithful representation of the shape of the data. More precisely:
>
> - (M) The sets in the cover should be **small** with respect to the **size** of the dataset.
>
> - (G) The sets in the cover should be **regular** with respect to the **geometry** of the data.
>
> - (T) The nerve of the cover should be a **faithful** representation of the **topology** of the data.

---

We formalize condition (M) using measure theory, condition (G) using geometry, and condition (T) using topology, specifically, homology (Appendix A.2).

## 1.4. Cover learning vs. clustering

The clustering problem can be understood as the zero-dimensional case of the cover learning problem, in which the relevant geometry of the data is entirely determined by the connected components, with no relevant topological information above dimension zero. Although the cover learning problem cannot be completely reduced to the clustering problem, our proposed method does rely on clustering for initialization; see Section 4.3.

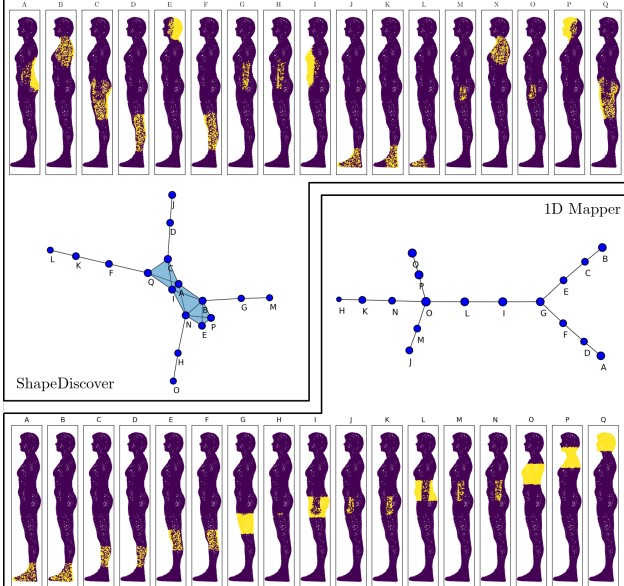

*Figure 1.* Two covers of a human 3D point cloud, obtained with ShapeDiscover and 1D Mapper, and the nerve of the covers. 1D Mapper's nerve is a graph, with no topological information above dimension 1. ShapeDiscover's nerve is a 2-dimensional simplicial complex, reflecting the fact that the human shape is hollow.

## 1.5. Contributions

We describe our four main contributions.

1. We identify the cover learning problem as a general purpose unsupervised learning problem. We observe that several methods from TDA are in essence cover learning algorithms, and review their strengths and limitations (Section 2).

2. We give a formal interpretation of the cover learning problem from the viewpoint of geometry and topology (Goal 2), and derive a principled loss function for cover learning (Optimization Problem 1), in the idealized scenario where the data consists of a Riemannian manifold. Our main theoretical result (Theorem 3.2) shows that the terms in our loss function can be either computed or bounded from above effectively with standard losses.

3. We propose practical estimators for the terms in our loss function, in the case where the space is a weighted graph, and show that optimization is feasible using known optimization tools, including (graph) neural networks and topological persistence optimization (Section 4).

4. We provide an implementation of ShapeDiscover (Scoccola & Lim, 2025), a cover learning algorithm based on our theory, and showcase it on two sets of experiments:

a quantitative one on topological inference, and a qualitative one on large-scale topology visualization. In the first case, ShapeDiscover learns topologically correct simplicial complexes, on synthetic and real data, of smaller size than those obtained with previous topological inference approaches. In the second, we argue that ShapeDiscover represents the large-scale topology of real data better, and with more intuitive parameters, than previous TDA algorithms that fit the cover learning framework.

### 1.6. Structure of the paper

In Section 2, we overview related work, specifically algorithms in TDA that fit the framework of cover learning. In Section 3, we develop theory for cover learning based on optimization, which we use in Section 4 to design a cover learning algorithm. Section 5 contains computational examples showcasing our methods. Appendix A has background, and other appendices contain details about theory and experiments.

## 2. Related work: cover learning in TDA

Although the concept of cover learning algorithm is introduced in this paper (Section 1), several unsupervised learning algorithms in TDA are in essence cover learning algorithms. Here, we overview some of these. For other related work, see Appendix E.

**Cover learning algorithms with a filter function.** To the best of our knowledge, 1D Mapper (Singh et al., 2007) is the first cover learning algorithm designed with the nerve construction in mind (see Algorithm 1 for the cover learning algorithm underlying 1D Mapper). The advantages of 1D Mapper are its simplicity and flexibility: different choices of the function $f : X \to \mathbb{R}$, known as the *filter function*, can provide widely different outputs, reflecting the structure of the data as seen by the filter function. We now describe four of 1D Mapper's main limitations:

1. *Dependence and choice of filter.* The filter function can be hard to choose if there are no obvious candidates. Moreover, to the best of our knowledge, there are no general methodologies for understanding the dependence of 1D Mapper on the filter function.

2. *Dependence and choice of cover of range of filter.* The initial cover $\{I_i\}_{i=1}^k$ of $\mathbb{R}$ (or, equivalently, of the range $[\min f, \max f]$ of the filter function) is also difficult to choose. This requires choosing the number of intervals $k$, as well as their endpoints.

3. *Dependence and choice of clustering algorithm.* The choice of clustering algorithm $C_\theta$ and its parameter(s) $\theta$ also has a big impact on the output, and is a difficult choice, as is familiar from cluster analysis.

---

**Algorithm 1** 1D Mapper cover learning algorithm

---

**Input:** Data $X$, function $f : X \to \mathbb{R}$, clustering algorithm $C_\theta$, parameter(s) $\theta$ for $C_\theta$, cover $\{I_i\}_{i=1}^k$ of $\mathbb{R}$

Take pullback cover $\{f^{-1}(I_i)\}_{i=1}^k$ of $X$

Let $\mathcal{U}_i := C_\theta(f^{-1}(I_i))$ for $1 \le i \le k$

**Return** The union $\bigcup_{i=1}^k \mathcal{U}_i$

---

**Algorithm 2** Ball Mapper cover learning algorithm

---

**Input:** Data $X$, $\varepsilon > 0$

Build an $\varepsilon$-net $\{y_i\}_{i=1}^k$ of $X$

**Return** The cover $\{B(y_i, \varepsilon)\}_{i=1}^k$

---

4. *Unsuitability for higher-dimensional topological inference.* The nerve of the cover produced by 1D Mapper is 1-dimensional, and does not contain higher-dimensional information. We elaborate on this in Appendix F.

Many strategies have been proposed to overcome the above limitations; here we overview some of these. In (Chalapathi et al., 2021), limitation (2) is addressed by automatically refining the initial cover of the range of the filter function using information-theoretic criteria; limitation (3) it also partially addressed by automatically searching for parameters for the clustering algorithm. Similarly, (Deb et al., 2018; Bui et al., 2020; Alvarado et al., 2024) aim at making the choice of initial cover of the range easier and more robust, by, e.g., using variations in data density, while (Ruscitti & McInnes, 2024) exploits variations in data density to improve the pullback cover directly. Multiscale Mapper (Dey et al., 2016) addresses limitation (2), at the expense of producing a one-parameter family of covers of the data (rather than a single cover), which can be analyzed using persistence. Limitation (4) can be addressed, as proposed in the original paper (Singh et al., 2007), by allowing the filter function to take values in a higher-dimensional Euclidean space, which renders the choice of the initial cover $\{I_i\}_{i=1}^k$ of the codomain of the filter function $\mathbb{R}^n$ significantly more delicate; we refer to this as multidimensional Mapper.

Recently, and closest to this paper, Differentiable 1D Mapper (Oulhaj et al., 2024) was introduced to address (1) by using optimization to automatically search for a filter function; this comes at the expense of having to choose a loss function and a parametric family of filter functions.

Many variations of Mapper exist, and Mapper in general has been shown to recover meaningful geometric structure in applications (Nicolau et al., 2011; Rizvi et al., 2017). However, we believe that the existence of a suitable filter function is a strong assumption, and that it is worth considering alternative solutions to the cover learning problem.

**Cover learning without a filter function.** The category of cover learning algorithms which do not rely on a filter function (implicitly or explicitly) is remarkably small. To the best of our knowledge, there is a single TDA cover learning algorithm in this category: Ball Mapper (Dłotko, 2019a;b), described in Algorithm 2. Consistency results can be proven for Ball Mapper, which imply that, contrary to cover learning algorithms based on 1D filter functions, Ball Mapper can be used to do higher-dimensional topological inference.

Ball Mapper's main limitation is its reliance on a global distance scale $\varepsilon > 0$, which must be chosen by the user, does not allow for variations in density, and is susceptible to the curse of dimensionality. Of note is the fact that Ball Mapper is essentially equivalent to the Witness complex (De Silva & Carlsson, 2004) with parameter $v = 0$ (Appendix G).

In conclusion, the only two TDA cover learning algorithms that do not require the user to provide a filter function are Ball Mapper and Differentiable Mapper, and the latter still requires a parametric family of filter functions.

# 3. Theory

For the purposes of deriving a sensible loss function for cover learning, let us first interpret Goal 1 in the case where the data consists of a compact $d$-dimensional Riemannian manifold $\mathcal{M}$. In order to make optimization feasible, we later extend the optimization space from that of covers to that of fuzzy covers (a certain space of function on $\mathcal{M}$).

**Short topology recap.** We need three standard notions from topology: homology, homology-good covers, and the nerve theorem; references are given in Appendix A, but let us describe the basic ideas here, for convenience. The *homology* construction takes as input a topological space $X$, and returns vector spaces $\mathsf{H}_i(X)$ for each $i \in \mathbb{N}$, in such a way that $\dim(\mathsf{H}_i(X))$ counts the number of $i$-dimensional holes in $X$; for example, $\mathsf{H}_0(X)$ counts connected components, while $\mathsf{H}_1(X)$ counts essential loops. A *homology-good* open cover of a topological space is a cover $\{U_i\}_{i \in I}$ such that, for every $J \subseteq I$, the reduced homology of $\bigcap_{j \in J} U_j$ is zero. The *nerve theorem* says that if $\mathcal{U}$ is a homology-good open cover of a topological space $X$, then the homology of the nerve $\mathsf{Ner}(\mathcal{U})$ is the same as that of $X$.

## 3.1. Formal goal

Condition (M) in Goal 1 is formalized by requiring the cover elements to have small measure, condition (G) by asking that the cover elements have boundary of small measure, while condition (T) is formalized by asking the cover to be homology-good, which ensures that the nerve recovers the correct homology, by the nerve theorem.

---

*Goal* 2. Find open cover $\{U_i\}_{i \in I}$ of $\mathcal{M}$ such that:

- (M) $\mathcal{H}^d(U_i)$ is small for all $i$.

- (G) $\mathcal{H}^{d-1}(\partial U_i)$ is small for all $i$.

- (T) $\{U_i\}_{i \in I}$ is a homology-good cover of $\mathcal{M}$.

Where $\mathcal{H}^\ell$ is the $\ell$-dimensional Hausdorff measure.

---

## 3.2. Optimization over spaces of covers

Solving Goal 2 directly using optimization is difficult, as the set of covers of a space does not have a natural smooth structure (even worse, when the space is finite, as is the case for finite data, the space of covers is discrete). We thus turn our focus to a larger space: that of fuzzy covers, which is a space of functions, and which is easier to parametrize.

### 3.2.1. FUZZY COVERS

Observe that a $k$-element cover of a set, i.e., a cover of the form $\{U_i\}_{i=1}^k$, is the same as a function $g : X \to \{0,1\}^k$ such that $\max_{1 \le i \le k} g_i(x) = 1$ for all $x \in X$. Here, the $\max$ operation is ensuring that every element belongs to some cover element. Indeed, the equivalence is given by mapping a cover $\{U_i\}_{1 \le i \le k}$ to the vector $(\chi_{U_1}, \ldots, \chi_{U_k})$ : $X \to \{0,1\}^k$ of indicator functions.

In order to turn the space of covers into a continuous space, we replace the two-element set $\{0,1\}$ by the unit interval $[0,1]$, and define

$$\Gamma^{k-1} := \left\{ p \in [0,1]^k : \max_{1 \le i \le k} p_i = 1 \right\} \subseteq \mathbb{R}^k. \quad (1)$$

The exponent $k-1$ in $\Gamma^{k-1}$ represents the topological dimension of the space. This notation is consistent with the usual convention that denotes the standard simplex with $k$ vertices by $\Delta^{k-1}$ (see also Section 4.2.1).

**Definition 3.1.** A $k$-element *fuzzy cover* of a set $X$ is a function $X \to \Gamma^{k-1}$.

In order to produce a single cover out of a fuzzy cover, one can threshold it, as follows. Given any function $f : X \to [0,1]$ and a threshold $\lambda \in [0,1]$, consider the *suplevel set* $\{f > \lambda\} := \{x \in X : f(x) > \lambda\}$. Then, given a fuzzy cover $g : X \to \Gamma^{k-1}$ and $\lambda \in [0,1]$, we can threshold the fuzzy cover $g$ to obtain a cover of $X$, as follows:

$$\{g > \lambda\} := \big\{ \{g_i > \lambda\} \big\}_{1 \le i \le k}.$$

The fact that $\{g > \lambda\}$ is indeed a cover of $X$ follows immediately from the $\max$-condition in Equation (1).

Since this will be of use later, let us remark here that a fuzzy cover $g : X \to \Gamma^{k-1}$ gives rise to a filtered simplicial complex $\{\mathsf{Ner}(\{g > \lambda\})\}_{\lambda \in [0,1]}$ (see Appendix D).

### 3.2.2. RANDOM COVERS FROM FUZZY COVERS

Suppose that $g : X \to \Gamma^{k-1}$ is a fuzzy cover of $X$. By sampling a threshold $\lambda \in [0, 1]$ uniformly at random, and considering $\{g > \lambda\}$, we get a random cover of $X$, so each fuzzy cover $g$ induces a probability measure $\mu_g$ on the space of covers of $X$. Formally, the probability measure $\mu_g$ is the pushforward of the uniform measure on $[0, 1]$ along the function $\lambda \mapsto \{g > \lambda\}$ (see Appendix B.3 for details).

Random covers in the context of TDA were first considered in (Oulhaj et al., 2024), with the similar goal of producing a cover of a dataset via optimization (Section 2). Their methodology, however, follows the 1D Mapper pipeline, and relies on filter functions.

### 3.3. The idealized loss function

**Loss function for a Riemannian manifold.** Our idealized loss function for fuzzy cover learning is obtained by taking the expected value of the conditions in Goal 2 for a random cover distributed according to the probability measure $\mu_g$ corresponding to a fuzzy cover $g$.

---

*Optimization Problem* 1.

$$\min_{\substack{g: \mathcal{M} \to \Gamma^{k-1} \\ \text{smooth}}} \alpha_M \cdot \mathsf{M}(g) \; + \; \alpha_G \cdot \mathsf{G}(g) \; + \; \alpha_T \cdot \mathsf{T}(g)$$

$$\mathsf{M}(g) := \sum_{i=1}^{k} \mathbb{E}\Big( \mathcal{H}^d(U_i) \Big)^2$$

$$\mathsf{G}(g) := \sum_{i=1}^{k} \mathbb{E}\Big( \mathcal{H}^{d-1}(\partial U_i) \Big)^2$$

$$\mathsf{T}(g) := \mathbb{P}\Big( \{U_i\}_{i=1}^{k} \text{ is not homology-good} \Big)$$

$\{U_i\}_{i=1}^{k}$ random cover with probability measure $\mu_g$

---

Next is our main theoretical result, which allows us to minimize the loss terms in Optimization Problem 1 in practice.

**Theorem 3.2.** *If $\mathcal{M}$ be a compact Riemannian manifold and $g : \mathcal{M} \to \Gamma^{k-1}$ a smooth fuzzy cover, then*

$$\mathsf{M}(g) = \sum_{i \in I} \|g_i\|_1^2$$

$$\mathsf{G}(g) = \sum_{i \in I} \|\nabla g_i\|_1^2$$

$$\mathsf{T}(g) \le \sum_{J \subseteq I} \left\| \min_{j \in J} g_j \right\|_{\mathsf{H}}^2,$$

*where $I = \{1, \ldots, k\}$. Moreover, in the third case, the left-hand side is zero if and only if the right-hand side is.*

In the statement of Theorem 3.2, if $f : X \to [0, 1]$ is a function on a topological space, we define $\|f\|_{\mathsf{H}} = \int_0^1 \dim \overline{\mathsf{H}}(\{f > r\}) \, \mathrm{d}r$, the integral of the dimension of the reduced homology of the suplevel sets of $f$. This quantity is also known as the length of the barcode or total persistence; see Appendix A.3.4 for details. The important point here is that this quantity can be estimated and optimized for in practice thanks to the theory of topological persistence optimization, effectively turning the nerve theorem into a loss function. See Appendix B.1 for a proof of Theorem 3.2.

**Regularization.** For theoretical and practical reasons, we add an $L^2$-regularization term to Optimization Problem 1, in the form of $\mathsf{R}(g) := \sum_{i \in I} \|\nabla g_i\|_2^2$. From the theoretical point of view, we believe that this is necessary to guarantee existence of minima; from the practical point of view, this deals with the curse of dimensionality inherent in optimizing over a space of functions on the data.

**Homology-good covers up to a dimension.** In practice, one might be interested in a cover whose nerve recovers the homology of the underlying space only up to dimension $\ell \in \mathbb{N}$. This can be accomplished using the notion of $\ell$-homology-good cover (Definition A.4) and a refined version of the nerve theorem (Theorem A.5). The loss $\mathsf{T}$ can also be relaxed accordingly to get a loss $\mathsf{T}_\ell$, which only accounts for homology up to degree $\ell$, and the upper bound of Theorem 3.2 can also be relaxed in this case; this is described in Appendix B.2. The upshot is that, if only homology up to a certain dimension is relevant, then a less computationally intensive topological loss can be used.

## 4. Practice

Our practical approach to cover learning relies on three main considerations: estimating the loss of Optimization Problem 1 (Section 4.1), finding a suitable parametrization of the space of fuzzy covers (Section 4.2), and computing a good initialization (Section 4.3). Other practical considerations are required to make optimization feasible and efficient (Section 4.4). Building on these, we describe the ShapeDiscover cover learning algorithm in Section 4.5.

### 4.1. Loss estimation

Let $X$ be the input data (seen as a set), and let $G$ be a weighted graph with $X$ as its set of vertices, and with weight $w(x, y)$ for $(x, y) \in G$ representing relationship strength between $x$ and $y$. Let $g : X \to [0, 1]^k$ (for example, $g$ could be a fuzzy cover). Based on Theorem 3.2, we define the following estimators for the losses:

$$\widehat{\mathsf{M}}(g) = \eta_M \cdot \sum_{i=1}^{k} \left( \sum_{x \in X} g_i(x) \right)^2$$

$$\widehat{\mathsf{G}}(g) = \eta_G \cdot \sum_{i=1}^{k} \left( \sum_{(x,y) \in G} w(x,y) \cdot |g_i(x) - g_i(y)| \right)^2$$

$$\widehat{\mathsf{T}}_0(g) = \eta_T \cdot \sum_{i=1}^{k} \|g_i\|_{\mathsf{H}_0}^2$$

$$\widehat{\mathsf{R}}(g) = \eta_R \cdot \sum_{i=1}^{k} \sum_{(x,y) \in G} w(x,y) \cdot |g_i(x) - g_i(y)|^2,$$

where $\eta_M, \eta_G, \eta_T, \eta_R$ are normalization constants (depending only on the graph) for the purpose of having the losses be in similar ranges; see Appendix H.1.

For the topological loss, we use the relaxation which only considers 0-dimensional homology (Appendix B.2), since 0-dimensional homology is more efficient to compute than higher-dimensional homology, and still performs well in the examples (Section 5.1).

## 4.2. Parametric fuzzy covers

In order to make our approach flexible enough to be combined with standard machine learning models, we first describe a way of turning any parametric vector-valued model $f_\theta : X \to \mathbb{R}^k$ on the input data $X$ into one of the form $g_\theta : X \to \Gamma^{k-1}$, thus parametrizing the space of fuzzy covers on the data. See Section 4.2.2 for possible initial models $f_\theta$, including neural networks and graph neural networks.

### 4.2.1. PARAMETRIZING FUZZY COVERS WITH SOFTMAX

Let $p \in [1, \infty]$, and define

$$\Delta_p^{k-1} = \left\{ x \in [0,1]^k : \|x\|_p = 1 \right\},$$

so that, in particular, $\Delta_\infty^{k-1} = \Gamma^{k-1}$, and $\Delta_1^{k-1}$ is the standard $(k-1)$-dimensional simplex. A function $h : X \to \Delta_1^{k-1}$ is called a *partition of unity*, since it consists of functions $h_1, \dots, h_k : X \to [0,1]$ such that $\sum_i h_i \equiv 1$. Note that, by definition of the standard softmax function, we have $\mathsf{softmax} : \mathbb{R}^k \to \Delta_1^{k-1}$. Thus, given any parametric model $f_\theta : X \to \mathbb{R}^k$, we obtain a parametric partition of unity by simply postcomposing with $\mathsf{softmax}$.

Now, note that for every $p \in [1, \infty]$, there is a function $\pi_p : \Delta_1^{k-1} \to \Delta_p^{k-1}$ given by normalization, i.e., by mapping $x$ to $x/\|x\|_p$. So if $f_\theta : X \to \mathbb{R}^k$ is a parametric model, we get a parametric fuzzy cover as follows:

$$\pi_\infty \circ \mathsf{softmax} \circ f_\theta : X \longrightarrow \Delta_\infty^{k-1} = \Gamma^{k-1}.$$

### 4.2.2. PARAMETRIC MODELS

Depending on the type of data $X = \{x_1, \dots, x_n\}$, there are different choices for $f_\theta$:

- The simplest option to let $\theta$ be an $n$-by-$k$ matrix, and to set $f_\theta(x_i)_j = \theta_{i,j}$.

- If $X \subseteq \mathbb{R}^N$, then we can let $\varphi_\theta : \mathbb{R}^N \to \mathbb{R}^k$ be a neural network and set $f_\theta(x_i) = \varphi_\theta(x_i)$.

- If $X$ consists of the vertices of a graph, then we can let $\varphi_\theta : (\mathbb{R}^N)^X \to (\mathbb{R}^k)^X$ be a graph neural network, let $V \in (\mathbb{R}^N)^X$ consist of a data-dependent vector (such as the first $N$ eigenfunctions of the graph Laplacian), and set $f_\theta(x_i) = \varphi_\theta(V)(x_i)$.

## 4.3. Initialization

In order to make optimization fast and reliable, it is necessary to use a good initialization. Any clustering of a set is in particular a cover, and we propose to use a geometrically meaningful clustering (obtained by, e.g., $k$-means or spectral clustering) as initial fuzzy cover.

## 4.4. Other implementation details

### 4.4.1. SMOOTH PARAMETRIC FUZZY COVERS

As described in Section 4.2, if $f_\theta : X \to \mathbb{R}^k$ is a parametric vector-valued model, then $\pi_\infty \circ \mathsf{softmax} \circ f_\theta : X \to \Delta_\infty^{k-1} = \Gamma^{k-1}$ is a parametric fuzzy cover. However, the operation $\pi_\infty$ given by normalization by the $\infty$-norm is not smooth. We thus propose to replace $\pi_\infty$ with $\pi_p$ for some $p \in [1, \infty)$ during optimization, and to only use $\pi_\infty$ to obtain the final fuzzy cover; see Algorithm 3. We expect larger values of $k$ to require larger values of $p$, but this approximation does not seem to influence results significantly, and we let $p = 5$ in all experiments.

### 4.4.2. TOPOLOGICAL OPTIMIZATION

In order to optimize the loss $\widehat{\mathsf{T}}_0$ we use the topological optimization with big steps of (Nigmetov & Morozov, 2024); see Appendix A.3.4.

## 4.5. ShapeDiscover

See Algorithm 3 for a high-level description of the procedure; here we give details about the subroutines. For details about the software we use see Appendix I.1, and for computational complexity see Appendix C.

SUBROUTINE NeighborhoodGraph

The simplest approach is to use a nearest neighbor graph, with weights equal to 1. This worked well in most of our experiments. However, to make the method more robust, our implementation uses the weighted neighborhood graph of the UMAP algorithm (see Section 3.1 of (McInnes et al., 2018)); this is what we use in the computational examples.

---

**Algorithm 3** ShapeDiscover fuzzy cover learning algorithm

**Input:** Point cloud $X \subseteq \mathbb{R}^N$

**Parameters:** `n_cov` $\in \mathbb{N}$, `n_neigh` $\in \mathbb{N}$, `reg` $> 0$

**Optimization parameters:** `lr`, `n_epoch`, $p \in [1, \infty)$

$G := \text{NeighborhoodGraph}(X, \text{n\_neigh})$

$g := \text{FuzzyCoverInitialization}(G, \text{n\_cov})$

$h := \text{ParametricPartitionOfUnity}()$

$\theta' := \text{InitializeParametricModel}(h, g)$

$\mathcal{L}(\theta) := \left( \widehat{\mathsf{M}} + \text{reg} \cdot \widehat{\mathsf{G}} + \widehat{\mathsf{T}} + \text{reg} \cdot \widehat{\mathsf{R}} \right) (\pi_p \circ h_\theta)$

$\theta := \text{GradientDescent}(\mathcal{L}, \text{n\_epoch}, \text{lr}, init = \theta')$

**Return** $\pi_\infty \circ h_\theta$

---

SUBROUTINE FuzzyCoverInitialization

We use spectral clustering as initialization: We compute the first `n_cov` eigenvalues of the normalized Laplacian of $G$, and apply $k$-means with $k = \text{n\_cov}$ to get a clustering of $X$, which we then simply interpret as a fuzzy cover, as described in Section 4.4.

SUBROUTINE ParametricPartitionOfUnity

We use the simplest of the approaches described in Section 4.2: we let $\varphi$ be a $|X|$-by-`n_cov` real matrix, and let $f_\theta : X \to \mathbb{R}^k$ be defined by $f_\theta(x_i)_j = \theta_{i,j}$. Then, the parametric partition of unity is defined as softmax $\circ f_\theta : X \to \Delta_1^{k-1}$.

SUBROUTINE InitializeParametricModel

Since we use a very simple model, initialization is straightforward, and it amounts to setting $\theta'_{i,j} \leftarrow g(x_i)_j$.

SUBROUTINE GradientDescent

We train for `n_epoch` (or convergence) using an Adam optimizer with learning rate `lr` and default parameters.

### 4.6. Output and parameter selection

We elaborate on how to use ShapeDiscover's output, and on how we choose its parameters in the experiments. See also Appendix I.9 for ablation study and parameter and noise sensitivity analyses.

#### 4.6.1. OUTPUT

The output of Algorithm 3 is a fuzzy cover. Using the nerve construction (Definition A.1), the fuzzy cover induces a filtered simplicial complex (Appendix D), indexed by $[0, 1]$. Given any $\lambda \in [0, 1]$, the filtered simplicial complex can be thresholded to a single simplicial complex.

#### 4.6.2. DEFAULT PARAMETERS

In all experiments we fix the following default parameters: number of neighbors for the neighborhood graph `n_neigh` $= 15$, regularization parameter `reg` $= 10$, number of iterations for gradient descent `n_epoch` $= 500$, learning rate for gradient descent `lr` $= 0.1$, and approximation parameter for fuzzy cover $p = 5$ (Section 4.4).

When a filtered simplicial complex is needed, we reindex the output of Algorithm 3 by $-\log$ to have it be indexed by $[0, \infty)$ as standard topological inference algorithms (Appendix D). When a simplicial complex is needed, we threshold with $\lambda = 0.5 \in [0, 1]$, by default.

## 5. Experiments

See appendix for details about software (Appendix I.1), datasets (Appendix I.2), computation time (Table 4), computational complexity (Appendix C), as well as an ablation study and parameter and noise sensitivity analyses (Appendix I.9).

We use ShapeDiscover with default parameters in all experiments (Section 4.6). We only vary the maximum cover size (`n_cov` $\in \mathbb{N}$) and threshold $\lambda \in [0, 1]$ (but only in MNIST and C. elegans data, otherwise default $\lambda = 0.5$ is used).

### 5.1. Topological inference

A. EFFICIENT TOPOLOGICAL REPRESENTATION

We compare ShapeDiscover to the standard approaches to topological inference on four datasets: a synthetic 2-sphere, a synthetic 3-sphere, a sample of the surface of a human (which is topologically equivalent to a 2-sphere), and the dynamical system video data from (Lederman & Talmon, 2018) (which is topologically equivalent to a 2-torus). We do not use Mapper since current implementations are not suitable for higher-dimensional topological inference (Appendix I.5). Each algorithm takes as input a number of vertices, and the goal is to find the smallest number of vertices that achieves a certain homology recovery quotient, quantified as the proportion of filtration values at which the correct homology is recovered (see Appendix I.4 for the definition). For more details; see Appendix I.3. Results are in Tables 1 and 2. ShapeDiscover performs best, followed by Witness ($v = 1$).

B. TOROIDAL TOPOLOGY IN NEUROSCIENCE

In Appendix I.6, we describe an experiment on the toroidal topology of population activity in grid cells in the dataset of (Gardner et al., 2022). In it, ShapeDiscover is able to learn this topology using a simplicial complex with just 15 vertices (see Figure 6), a task that Vietoris–Rips is not able to accomplish with a simplicial complex with 5000 vertices.

## 5.2. Visualization of large-scale structure

For plotting conventions, see Appendix I.7.

### C. HUMAN AND OCTOPUS

We compare ShapeDiscover to 1D Mapper (height as filter function), Ball Mapper, and Differentiable Mapper on human and octopus datasets (Figure 7). In Figures 1 and 2, ShapeDiscover recovers the large-scale topology, including the fact that the figures are hollow, with a small number of vertices, while 1D Mapper and Differentiable Mapper recover the 1-dimensional structure, but not the higher-dimensional one (Section 2). In Figures 11 and 12, Ball Mapper has a main parameter ($\varepsilon$) that is hard to tune, and has a hard time recovering the correct topology.

In the final two examples, we are interested in cover learning without extra information (e.g., filter function), so we experiment with Ball Mapper and Differentiable Mapper.

### D. HANDWRITTEN DIGIT DATASETS

We run ShapeDiscover and UMAP on a small handwritten digits dataset and on MNIST. In Figures 4 and 15 UMAP and ShapeDiscover capture similar structure, corresponding well to digits and their similarities. We did not obtain meaningful results with other algorithms (Figure 14 and Appendix I.8).

### E. FLARE STRUCTURE IN BIOLOGY

We use the data from (Packer et al., 2019), consisting of the gene expressions of cells of the C. elegans worm, and known to have flare structure corresponding to single-cell trajectories. We project the data to 2 dimensions using UMAP, which clearly identifies the flare structure of 16 cell types (1–16); see Figure 16 and Table 3. ShapeDiscover identifies the same flare structure (Figure 5). We were not able to obtain meaningful results with other algorithms (Figure 17 and Appendix I.8).

## 6. Conclusions

Motivated by the problem of representing the large-scale topology of geometric data for the purposes of efficient topological inference and visualization, we have developed a theory for cover learning based on optimization, as well as a practical implementation of these ideas with easy-to-tune parameters. We demonstrated that cover learning can improve upon the two main TDA methodologies with a single approach, by effectively addressing their main shortcomings: the sometimes prohibitive size of geometric complexes, and the difficulty in tuning and the lack of higher dimensional information in Mapper graphs.

We believe that cover learning is relevant to unsupervised learning, and to clustering and dimensionality reduction specifically. This paper is only a first step towards the development of efficient and robust cover learning methods. We now elaborate on the limitations of our approach, and give further motivation for future research directions.

*Robust cover learning.* Covers produced by cover learning algorithms will often contain spurious intersections, which can render visualization difficult, and lead to incorrect topology recovery. This motivates the developing of robust cover learning, and of reliable and interpretable layout algorithms for cover visualization.

*Consistency.* Standard topological inference algorithms come with topology recovery guarantees, e.g., (Boissonnat et al., 2018; Kim et al., 2020), and here we have not provided such guarantees for optimization-based cover learning procedures. As an avenue for future work, we observe that the nerve theorem can be used to formulate computable conditions for covers which, when satisfied, guarantee topological correctness of the cover, regardless of how it was obtained.

*Efficiency.* Topological persistence optimization is slow for a few reasons, including the fact that the whole dataset is required at each iteration step, precluding the usage of mini-batch during optimization. It is an open problem to find methods for topological regularization (i.e., for enforcing trivial topology), which can be computed in small batches of the data. It may be possible to develop such methods using probabilistic, sample-based algorithms for topological estimation.

*Other cover learning algorithms.* Cover learning being a general unsupervised learning problem (like clustering), there is no single, ultimate cover learning algorithm, and other approaches should be considered. In particular, one could simplify our pipeline, as suggested by the ablation study (Appendix I.9).

*Point cloud vectorization.* Point clouds can be vectorized by constructing a graph on the data and applying methods from graph machine learning. Cover learning can be used to build graphs on data that are of much smaller size than those built using classical methods such as $k$-nearest neighbors, effectively reducing the dimension of the learning problem.

*Persistence-based clustering.* A main difficulty with many clustering algorithms is that the final number of clusters needs to be chosen by the user. This difficulty can be addressed with methods from density-based clustering and persistence-based clustering (Campello et al., 2013; McInnes et al., 2017). Fuzzy cover learning naturally produces a filtered graph (Section 4.6.1), which can then be used as input to persistence-based clustering methods such as (Chazal et al., 2013; Rolle & Scoccola, 2024).

*Table 1.* Rows correspond to algorithms taking a number of vertices and outputting a filtered complex. Columns correspond to a dataset and a homology recovery quotient. Entries are the minimum number of vertices (left) and total number of simplices (right) needed to get the required homology recovery quotient; see Appendix I.3 for further details.

| Algorithm | 2-sph 0.25 | | 2-sph 0.5 | | 3-sph 0.25 | | 3-sph 0.5 | | human 0.01 | | human 0.05 | | dynam 0.01 | | dynam 0.05 | |
|---|---|---|---|---|---|---|---|---|---|---|---|---|---|---|---|---|
| S + Rips | 15 | 3.2e3 | 48 | 2.1e5 | 44 | 1.2e6 | 184 | 1.7e9 | 111 | 6.2e6 | 129 | 1.1e7 | 257 | 1.8e8 | 283 | 2.7e8 |
| S + Rips + DMS | - | 578 | - | 1.6e5 | - | 8.8e5 | - | OOM | - | OOM | - | OOM | - | OOM | - | OOM |
| S + Rips + edg.coll. | 15 | 1.5e3 | 48 | 1.6e5 | 44 | 8.0e5 | 184 | 1.5e9 | 111 | 1.5e3 | 129 | 2.0e3 | 257 | 2.4e7 | 283 | 3.5e7 |
| S + sparse Rips ($\varepsilon$=.5) | 17 | 3.2e3 | 48 | 1.6e5 | 44 | 1.2e6 | - | OOM | 111 | 4.4e5 | 129 | 6.5e5 | 305 | 6.2e5 | 283 | 4.9e7 |
| S + sparse Rips ($\varepsilon$=1) | 16 | 970 | 35 | 4.8e3 | 39 | 1.0e5 | - | OOM | 61 | 1.8e4 | 211 | 1.6e5 | 257 | 4.2e7 | 297 | 6.3e5 |
| S + Alpha | 10 | 83 | 15 | 159 | 14 | 355 | 34 | **1.5e3** | 85 | **1.8e3** | 261 | 6.7e3 | - | OOM | - | OOM |
| Witness ($v$=0) / MS BMapper | 15 | 1.9e3 | 48 | 2.1e5 | 44 | 1.2e6 | 184 | 1.7e9 | 46 | 1.8e5 | 193 | 5.7e7 | 52 | 2.9e5 | 60 | 5.2e5 |
| Witness ($v$=2) | 10 | 385 | 10 | 385 | 17 | 9.4e3 | **17** | 9.4e3 | 51 | 2.7e5 | 46 | 1.8e5 | 78 | 1.5e6 | 90 | 2.7e6 |
| Witness ($v$=1) | 6 | 56 | 6 | 56 | 8 | 218 | **17** | 9.4e3 | 26 | 1.8e4 | 26 | 1.8e4 | 62 | 6.0e5 | 82 | 1.8e6 |
| ShapeDiscover (v0.1) | **5** | **30** | **5** | **30** | **7** | **119** | 19 | 1.6e4 | **19** | 5.0e3 | **19** | **5.0e3** | **18** | **4.0e3** | **52** | **2.9e5** |

*Table 2.* We use the number of vertices required by ShapeDiscover to achieve the homology recovery quotients of Table 1 to run all main algorithms, and report the homology recovery quotient.

| Algorithm | 2-sph 5 | 3-sph 19 | human 19 | dynam 52 |
|---|---|---|---|---|
| S + Rips | 0 | 0.03 | 0 | < 0.01 |
| S + Alpha | 0.11 | 0.30 | < 0.01 | OOM |
| Witness ($v$=0) / MS BMapper | 0 | 0.08 | 0 | 0.04 |
| Witness ($v$=2) | 0 | **0.73** | 0 | 0 |
| Witness ($v$=1) | 0 | 0.65 | 0 | 0 |
| ShapeDiscover (v0.1) | **0.74** | 0.51 | **0.7** | **0.38** |

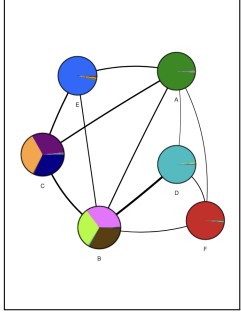

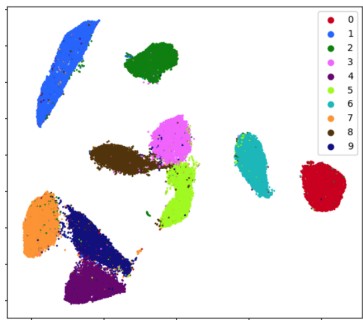

*Figure 4.* ShapeDiscover (n_cov = 15, $\lambda$ = 0.8) (left) and UMAP (right) on MNIST; see Figure 15 for ShapeDiscover's cover on the UMAP projection. The algorithms capture similar structure; a main difference is that ShapeDiscover isolates a large subset of the digit 1 (vertex I). Note that only 10 of the 15 elements of ShapeDiscover's cover are non-empty.

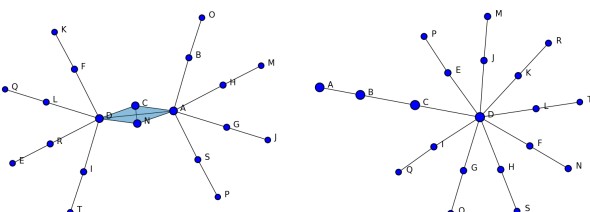

*Figure 2.* ShapeDiscover (n_cov = 20) (left) and 1D Mapper (right) on octopus dataset. See Figs. 8,9 for corresponding covers.

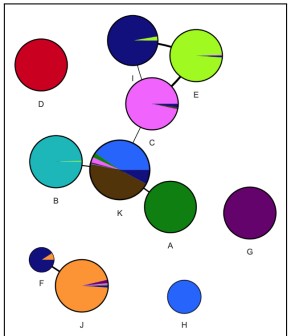
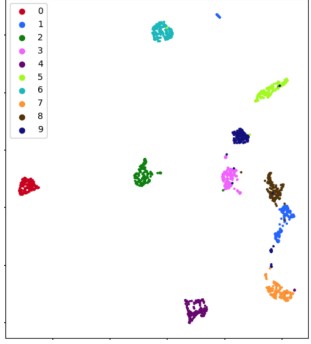

*Figure 3.* ShapeDiscover (n_cov = 15) (left) and UMAP (right) on the small digits dataset; see Figure 13 for ShapeDiscover's cover on the UMAP projection. We see that both algorithms capture essentially the same structure: most of the digits are clearly distinguished, except for 8 and 1. Note that only 11 of the 15 elements of ShapeDiscover's cover are non-empty.

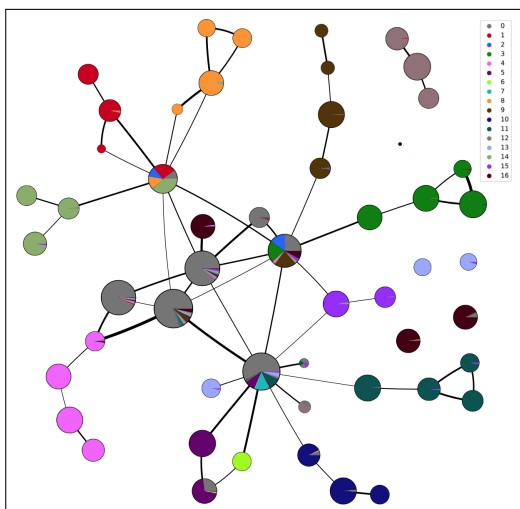

*Figure 5.* ShapeDiscover (n_cov = 100, $\lambda$ = 0.6) on C. elegans data, recovering most of the flare structure captured by UMAP (Figure 16 and Table 3); 53 elements in the cover are non-empty.

## Impact statement

This paper presents work whose goal is to advance the field of Machine Learning. There are many potential societal consequences of our work, none which we feel must be specifically highlighted here.

## Acknowledgments

We thank the anonymous reviewers at ICML 2025 for their helpful and knowledgeable comments. LS thanks Matt Piekenbrock for discussions about graph layout algorithms and Tatum Rask for help source datasets. HAH gratefully acknowledges funding from the Royal Society RGF\EA\201074, UF150238 and EPSRC EP/Y028872/1 and EP/Z531224/1. LS was supported by a Centre de Recherches Mathématiques et Institut des Sciences Mathématiques fellowship. UL was supported by Leverhulme Trust Research Project Grant RPG-2023-144, BMS Centre for Innovation and Translational Research Europe, and Korea Foundation for Advanced Studies. The authors were members of the UK Centre for Topological Data Analysis EPSRC grant EP/R018472/1.

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

# A. Background

## A.1. Simplicial complexes and nerve

A finite *simplicial complex* consists of a finite set $K_0$ of *vertices*, together with a set $K$ of subsets of $K_0$ with the property that $\{x\} \in K$ for every $x \in K_0$, and such that, for all $\sigma \subseteq \tau \in K$, we have $\sigma \in K$. The elements of $K$ are called *simplices*, the *dimension* of a simplex $\sigma \in K$ is $|\sigma| - 1 \in \mathbb{N}$, and the *dimension* of a simplicial complex is the maximum dimension of its simplices. For example, any finite, unoriented, simple graph $G$ can equivalently be seen as a 1-dimensional simplicial complex with 0-dimensional simplices the vertices of $G$, with 1-dimensional simplices the edges of $G$, and with no higher-dimensional simplices.

**Definition A.1.** Let $\mathcal{U} = \{U_i\}_{i \in I}$ be a cover of a set $X$. The *nerve* of $\mathcal{U}$, denoted $\mathsf{Ner}(\mathcal{U})$, is the following simplicial complex on $X$. The underlying set of $\mathsf{Ner}(\mathcal{U})$ is $I$ and the simplices of $\mathsf{Ner}(\mathcal{U})$ are sets $\sigma \subseteq I$ such that $\bigcap_{j \in \sigma} U_j \neq \emptyset$. The function $\mathsf{Ner}(\mathcal{U})_0 \to \mathsf{Parts}(X)$ maps $i \in I$ to $U_i$.

## A.2. Homology, homology-good covers, and nerve theorem

**Homology.** It is beyond the scope of this paper to give a formal definition of singular homology of topological spaces; instead, we recall the basic ideas here, and refer the reader to (Hatcher, 2002; Fomenko, 1994) for introductions to homology with extensive visual examples from the point of view of mathematics, to (Edelsbrunner & Harer, 2022; Oudot, 2015; Boissonnat et al., 2018; Ghrist, 2008) for introductions to homology from the point of view of applied topology, and to (Hensel et al., 2021; Coskunuzer & Akçora, 2024) for surveys of applications of topology to machine learning.

We fix a field, which we omit from notation. The *homology* construction takes as input a topological space $X$, and returns a vector space $\mathsf{H}_i(X)$ for each $i \in \mathbb{N}$, in such a way that $\dim(\mathsf{H}_i(X))$ counts the number of $i$-dimensional holes in $X$; for example, $\mathsf{H}_0(X)$ counts connected components, while $\mathsf{H}_1(X)$ counts essential loops. When referring to all homology groups at once, we write $\mathsf{H}(X)$.

As is standard, the homology $\mathsf{H}(K)$ of a simplicial complex $K$ is taken to be either the homology of its geometric realization or equivalently its simplicial homology.

Homology is functorial, which in particular means that every continuous map between topological spaces $X \to Y$ induces a linear morphism $\mathsf{H}_i(X) \to \mathsf{H}_i(Y)$ of homology vector spaces.

**Reduced homology.** The $i$th *reduced homology* of a topological space $X$ is the kernel of the linear map $\mathsf{H}(X) \to \mathsf{H}(*)$, where $X \to *$ is the only map from $X$ to the one-point topological space.

**Definition A.2.** An open cover $\mathcal{U} = \{U_i\}_{i \in I}$ of a topological is *homology-good* we have $\overline{\mathsf{H}}\left(\bigcap_{j \in J} U_j\right) = 0$, for every $J \subseteq I$.

**Nerve theorem.** The nerve theorem has many incarnations; see, e.g., (Bauer et al., 2023) for an overview of these. The version we use is the following:

**Theorem A.3** (Nerve theorem). *Let $\mathcal{U} = \{U_i\}_{i \in I}$ be an open cover of a topological space $X$. If $\mathcal{U}$ is homology-good, then $\mathsf{H}(X) \cong \mathsf{H}\left(\mathsf{Ner}(\mathcal{U})\right)$.* □

The above result is standard, and it follows, for example, from the refined version of the nerve theorem (Theorem A.5, below).

**Homology recovery up to dimension $\ell$.** In many applications of algebraic topology, only homology up to a certain dimension is relevant. For example, if one is interested in the connected components of a topological space, as is the case in clustering, only $\mathsf{H}_0$ is required. A general version of the nerve theorem, Theorem A.5 below, gives condition for recovery of homology only up to a certain dimension. We now abstract these conditions.

**Definition A.4.** Let $\mathcal{U} = \{U_i\}_{i \in I}$ be an open cover of a topological space $X$, and let $k \in \mathbb{N}$. The cover $\mathcal{U}$ is *$\ell$-homology-good* if $\overline{\mathsf{H}}_m\left(\bigcap_{j \in J} U_j\right) = 0$ for every $J \subseteq I$ with $|J| \leq \ell + 1$ and $0 \leq m \leq \ell - |J| - 1$.

We can now state the refined nerve theorem (for a proof, see, e.g., (Gillespie, 2022)).

**Theorem A.5.** *Let $\mathcal{U}$ be an open cover of a topological space $X$, and let $k \in \mathbb{N}$. If $\mathcal{U}$ is $k$-homology-good, then $\mathsf{H}_i(X) \cong \mathsf{H}_i\left(\mathsf{Ner}(\mathcal{U})\right)$ for every $0 \leq i \leq k$.*

### A.3. Topological persistence

For references, see (Edelsbrunner & Harer, 2022; Oudot, 2015; Boissonnat et al., 2018; Ghrist, 2008).

#### A.3.1. FILTERED SIMPLICIAL COMPLEXES

**Definition A.6.** A *filtered simplicial complex* $(K, f)$ consists of a simplicial complex $K$ together with a function $f : K \to \mathbb{R}$ from the simplices of $K$ to the real numbers, with the property that if $\sigma \subseteq \tau$ are simplices of $K$, then $f(\sigma) \leq f(\tau)$.

A filtered simplicial complex $(K, f)$ can equivalently be described as a sequence $\{K_r\}_{r \in \mathbb{R}}$ of simplicial complexes with the property that $K_r \subseteq K_s$ whenever $r \leq s$. To see this, we define $K_r = \{\sigma \in K : f(\sigma) \leq r\} \subseteq K$.

#### A.3.2. OTHER INDEXING POSETS

We remark that geometric complexes used in topological inference, such as the Vietoris–Rips complex (Appendix A.4) are often indexed by the non-negative real numbers. Another indexing poset of interest is $([0, 1], \geq)$, which indexes the filtered simplicial complexes associated to fuzzy covers; see Appendix D.

#### A.3.3. THE BARCODE

Applying homology in dimension $i$ to a filtered simplicial $(K, f)$ complex one gets a family of vector spaces $\{\mathsf{H}_i(K_r)\}_{r \in \mathbb{R}}$, and, by the functorial property of homology, we get linear maps $\mathsf{H}_i(K_r) \to \mathsf{H}_i(K_s)$ whenever $r \leq s \in \mathbb{R}$. This collection of vector spaces together with linear maps, often denoted as $\mathsf{H}_i(K, f)$, is known as a *persistence module*, and a fundamental result in persistence theory says that this persistence module can be described completely by a multiset of intervals in the real line, known as the *barcode* of the filtered simplicial complex, and often denoted $\mathcal{B}_i(K, f) = \{[a_\ell, b_\ell) \subseteq \mathbb{R}\}_{\ell \in L}$; see, e.g., (Ghrist, 2008).

The same considerations hold if one applies reduced homology instead of homology, in the sense that there exists a reduced barcode $\overline{\mathcal{B}}_i(K, f)$, in the form of a multiset of intervals of $\mathbb{R}$ which completely describes the persistence module obtained by applying reduced homology to a filtered simplicial complex $(K, f)$.

#### A.3.4. THE LENGTH OF THE BARCODE

If the $i$th dimensional barcode $\mathcal{B}_i(K, f)$ of a filtered simplicial complex $(K, f)$ is given by a multiset of intervals $\{[a_\ell, b_\ell) \subseteq \mathbb{R}\}_{\ell \in L}$, then its *length* (or *total persistence*) is given by the sum of the length of the intervals $|\mathcal{B}_i(K, f)| = \sum_{\ell \in L}(b_\ell - a_\ell)$.

When the simplicial complex is filtered by a bounded function, say $f : K \to [0, 1]$, then the length is a finite number $|\mathcal{B}_i(K, f)| \in \mathbb{R}$. Remarkably, the length of the barcode can be differentiated with respect to the filter function $f$ (Carrière et al., 2021), and this can be done efficiently using the methods in (Nigmetov & Morozov, 2024; Carrière et al., 2024).

It is a basic observation that the length of the barcode can be computed by integrating the dimension of the homology, that is: $|\mathcal{B}_i(K, f)| = \int_{-\infty}^{\infty} \dim \mathsf{H}_i(K_r)\, \mathrm{d}r$. As one might expect, if one integrates the dimension of the reduced homology, then one obtains the length of the reduced barcode.

### A.4. Topological inference and sparsification

Topological inference in a central topic in TDA; as its name suggests, the basic goal is to infer, from finite samples, topological properties (such as number of connected components or essential loops) of an underlying topological space. We refer the reader to books such as (Edelsbrunner & Harer, 2022; Boissonnat et al., 2018; Oudot, 2015) for an introduction. Arguably the most common construction for topological inference is the Vietoris–Rips complex, a particular kind of geometric complex that assigns, to any finite metric space $X$ (such as a point cloud) a filtered simplicial complex (i.e., a nested sequence of simplicial complexes, see Definition A.6) $\{\mathsf{VR}_\varepsilon(X)\}_{\varepsilon \geq 0}$ with vertices the elements of $X$, and a simplex on vertices $x_1, \ldots, x_k$ at scale $\varepsilon$ if $d_X(x_i, x_j) \leq \varepsilon$ for every $i, j \in \{1, \ldots, k\}$.

The number of simplices in the Vietoris–Rips complex grows extremely fast with the size of $X$, as it contains $\Theta(|X|^{d+1})$ $d$-dimensional simplices. For this reason, several sparsification techniques are commonly used, in particular: constructing the complex on a subsample of the data, reducing the complex using discrete Morse theory (Mischaikow & Nanda, 2013) and edge collapses (Boissonnat & Pritam, 2020), and approximations (Sheehy, 2013).

### A.5. Topological persistence optimization

There is extensive literature on the optimization of losses based on topology (Carrière et al., 2021; Leygonie et al., 2023; 2022; Hensel et al., 2021), with applications such as classification (Hofer et al., 2017; Poulenard et al., 2018; Hofer et al., 2020b; Horn et al., 2022; Zhao et al., 2020) representation learning (Hofer et al., 2019), and regularization (Chen et al., 2019; Hofer et al., 2020a; Scoccola et al., 2024; Moor et al., 2020). Of particular importance to this paper is the idea of topological optimization with big steps introduced in (Nigmetov & Morozov, 2024), which addresses the sparsity of gradient in vanilla topological persistence optimization, making gradient descent convergence significantly faster than previous approaches (for another, recent approach see (Carrière et al., 2024)). For more details, see Appendix A.3.

## B. Details about theory

### B.1. Computation of the loss function

**Lemma B.1.** *Let $f : \mathcal{M} \to [0, 1]$ be a smooth function on a $d$-dimensional Riemannian manifold $\mathcal{M}$. Then,*

$$\int_0^1 \mathcal{H}^d(\{f > \lambda\}) \, \mathrm{d}\lambda \;=\; \|f\|_1.$$

*Proof.* We simply compute

$$\begin{aligned}
\int_0^1 \mathcal{H}^d(\{f > \lambda\}) \, \mathrm{d}\lambda &= \int_{\lambda \in [0,1]} \int_{x \in \mathcal{M}} \mathbf{1}_{\{f > \lambda\}}(x) \, \mathrm{d}x \, \mathrm{d}\lambda \\
&= \int_{x \in \mathcal{M}} \int_{\lambda \in [0,1]} \mathbf{1}_{\{f > \lambda\}}(x) \, \mathrm{d}\lambda \, \mathrm{d}x \\
&= \int_{x \in \mathcal{M}} f(x) \, \mathrm{d}x = \|f\|_1,
\end{aligned}$$

where in the second equality we used Fubini's theorem. $\qquad\square$

**Lemma B.2.** *Let $f : \mathcal{M} \to \mathbb{R}$ be a smooth function on a Riemannian manifold. If $\nabla f$ does not vanish on $\{f = \lambda\}$, then $\partial\{f > \lambda\} = \{f = \lambda\}$.*

*Proof.* The left-to-right inclusion follows immediately from the fact that $f$ is continuous:

$$\partial\{f > \lambda\} = \overline{\{f > \lambda\}} \setminus \{f > \lambda\} \subseteq \{f \geq \lambda\} \setminus \{f > \lambda\} = \{f = \lambda\}.$$

For the right-to-left inclusion, it suffices to prove that $\{f = \lambda\} \subseteq \overline{\{f > \lambda\}}$. So, given $x \in \{f = \lambda\}$ we must show that every open neighborhood $U$ of $x$ intersects $\{f > \lambda\}$. Towards a contradiction, assume that there exists an open neighborhood $U$ of $x$ entirely contained in $\{f \leq \lambda\}$. Then $x$ is a maximum of $f$ restricted to $U$, and thus $\nabla f$ vanishes on $x$, a contradiction. $\quad\square$

**Lemma B.3.** *If $f : \mathcal{M} \to \mathbb{R}$ is a smooth function on a Riemannian manifold, then*

$$\int_0^1 \mathcal{H}^{d-1}\big(\partial\{f > \lambda\}\big) \, \mathrm{d}\lambda \;=\; \|\nabla f\|_1.$$

*Proof.* We claim that $\partial\{f > \lambda\} = \{f = \lambda\}$ for almost every $\lambda \in \mathbb{R}$; the statement of the lemma follows from this claim and the coarea formula, which in this case says that $\int_0^1 \mathcal{H}^{d-1}(\{f = \lambda\}) \mathrm{d}\lambda = \|\nabla f\|_1$ (see, e.g., Appendix A of (Jost & Li-Jost, 1998), for a proof of the coarea formula). So let us prove the claim. By Saard's theorem, the set of critical values of $f$ has measure zero. This implies that $\nabla f$ does not vanish on almost every level set $\{f = \lambda\}$, which implies our claim, thanks to Lemma B.2. $\qquad\square$

*Proof of Theorem 3.2.* Let $\varphi$ be a function which takes as input a cover and outputs a real number. Then, if a random cover $\mathcal{U}$ is distributed according to the probability measure $\mu_g$, we have that $\mathbb{E}(\varphi(\mathcal{U})) = \int_0^1 \varphi(\{g > \lambda\}) \, \mathrm{d}\lambda$. This is simply because $\mu_g$ is the pushforward of the uniform measure on $[0, 1]$ along the function $\lambda \mapsto \{g > \lambda\}$.

Then, the first equality follows from Lemma B.1, while the second follows from Lemma B.3.

For the inequality, note that $\mathbb{P}\left(\mathcal{U} \text{ is not homology-good}\right) = \mathbb{E}(\psi_{\mathcal{U}})$, where $\psi_{\mathcal{U}}$ is $0$ is $\mathcal{U}$ is good, and $1$ otherwise. By the considerations in the first paragraph, it follows that

$$
\begin{aligned}
\mathbb{P}\left(\mathcal{U} \text{ is not homology-good}\right) &= \int_0^1 \psi_{\{g > \lambda\}} \, \mathrm{d}\lambda \\
&\leq \int_0^1 \sum_{J \subseteq I} \dim \overline{\mathsf{H}}\left(\bigcap_{j \in J} \{g_j > \lambda\}\right)^2 \, \mathrm{d}\lambda \\
&= \int_0^1 \sum_{J \subseteq I} \dim \overline{\mathsf{H}}\left(\left\{\min_{j \in J} g_j > \lambda\right\}\right)^2 \, \mathrm{d}\lambda \\
&= \sum_{J \subseteq I} \left\| \min_{j \in J} g_j \right\|_{\mathsf{H}}^2,
\end{aligned}
$$

where in the inequality we simply used the fact that if a cover is not homology-good, then the dimension of the reduced homology of one of the intersections must be at least 1, by definition. Moreover, a cover is not homology-good if and only if the homology of one of the intersections is least 1 (also by definition), so the probability of $\mathcal{U}$ not being good is zero if and only if the upper bound is zero. $\qquad\square$

### B.2. Relaxation of the topological loss

Let $g : \mathcal{M} \to \Gamma^{k-1}$ be a fuzzy cover, and let $\mathcal{U} = \{U_i\}_{i=1}^k$ be a random cover distributed as $\mu_g$, as in Optimization Problem 1. Let $I = \{1, \ldots, k\}$. Fix a maximum homology dimension $\ell \in \mathbb{N}$ of interest. Then, the topological loss $\mathsf{T}(g) = \mathbb{P}\left(\{U_i\} \text{ is not homology-good }\right)$ can be relaxed to a loss $\mathsf{T}_\ell(g) = \mathbb{P}\left(\{U_i\} \text{ is not } \ell\text{-homology-good }\right)$, using Definition A.4. A derivation analogous to that in the proof of Theorem 3.2 shows that

$$
\mathsf{T}_\ell(g) \leq \sum_{\substack{J \subseteq I \\ |J| \leq \ell+1}} \sum_{m=0}^{\ell - |J| - 1} \left\| \min_{j \in J} g_j \right\|_{\mathsf{H}_m}^2,
$$

which, in the case where $g : K \to \Gamma^{k-1}$ is a fuzzy cover of a simplicial complex, leads us to define the following estimator

$$
\widehat{\mathsf{T}}_\ell(g) := \sum_{\substack{J \subseteq I \\ |J| \leq \ell+1}} \sum_{m=0}^{\ell - |J| - 1} \left\| \min_{j \in J} g_j \right\|_{\mathsf{H}_m}^2,
$$

In particular, we have

$$
\widehat{\mathsf{T}}_0(g) = \sum_{i \in I} \|g_i\|_{\mathsf{H}_0}^2,
$$

which only depends on the 1-skeleton of the simplicial complex, and thus makes sense for graphs.

### B.3. The probability measure associated to a fuzzy cover

If $X$ is a finite set, the space of $k$-element covers of $X$ is discrete and can be endowed with the discrete $\sigma$-algebra. If $g : X \to \Gamma^{k-1}$ is a fuzzy cover, then the map sending $\lambda \in [0, 1]$ to the cover $\{g > \lambda\}$ is measurable since $X$ only admits finitely many $k$-element covers, and for any $k$-element cover $\mathcal{U}$ of $X$, the set $\{r \in [0, 1] : \{g > r\} = \mathcal{U}\}$ is empty or a half-open interval. If $X$ is a compact metric space (e.g., a compact, connected Riemannian manifold), then its set of subsets can be endowed with the Hausdorff metric, which then gives a $\sigma$-algebra structure to the set of subsets, and thus to the set of $k$-element covers.

There is a more general theory of random sets, which deals with this type of situation; see, e.g., (Molchanov, 2017) or Chapter 18 (Aliprantis & Border, 2006). Here, we shall not go into details, since the functionals of the random covers we are interested in can be expressed as explicit integrals which do not involve random sets; see Theorem 3.2.

## C. Computational complexity

The bulk of ShapeDiscover's computation time is spent on optimization, that is, on executing the $\mathrm{GradientDescent}$ method in Algorithm 3, so let us analyze its computational complexity.

Let $X$ be the input point cloud. The neighborhood graph $G$ built on $X$ has size $|G| = O(\texttt{n\_neigh} \cdot |X|)$. From the loss estimation formulas (Section 4.1), we see that $\widehat{\mathsf{M}}$, $\widehat{\mathsf{G}}$, and $\widehat{\mathsf{R}}$ can be computed in $O(\texttt{n\_cov} \cdot |G|)$ time. A bit more interestingly, the loss $\widehat{\mathsf{T}}_0$, which involves the zero-dimensional persistence of a filtered graph, can be computed in $O(\texttt{n\_cov} \cdot |G| \cdot \log|G|)$ time, by sorting edges and using the elder rule (Edelsbrunner & Harer, 2010); see, e.g., Section 5.3.3 in (Rolle & Scoccola, 2024). The overall time complexity is thus:

$$O\left(\texttt{n\_epoch} \cdot \texttt{n\_cov} \cdot \texttt{n\_neigh} \cdot |X| \cdot \log|X|\right).$$

The space complexity depends on maintaining the neighborhood graph $G$ and the parameters of the model implementing the fuzzy cover. In the case of the model simply being an $|X|$-by-$\texttt{n\_cov}$ real matrix, as in Section 5, the space complexity is $O\left((\texttt{n\_cov} + \texttt{n\_neigh}) \cdot |X|\right)$.

## D. The filtered simplicial complex associated to a fuzzy cover

Let $X$ be a set, let $g : X \to \Gamma^{k-1} \subseteq [0,1]^k$ be a fuzzy cover, and let $I = \{1, \ldots, k\}$.

For each $\lambda \in (0,1]$ we define the simplicial complex $\mathsf{Ner}_\lambda(g) := \mathsf{Ner}(\{g > \lambda\})$, using the nerve construction, recalled in Section 1.

Note that, if $\lambda < \lambda'$, then the cover $\{g > \lambda'\}$ refines the cover $\{g > \lambda\}$, which means that there is an inclusion of simplicial complexes $\mathsf{Ner}_{\lambda'}(g) \subseteq \mathsf{Ner}_\lambda(g)$. In particular, $\{\mathsf{Ner}_\lambda(g)\}_{\lambda \in (0,1]}$ is a simplicial complex filtered by $((0,1], \geq)$, where the order on $(0,1]$ is the opposite of the standard one, since, as stated below, if $\lambda < \lambda'$, then the cover $\{g > \lambda'\}$ refines the cover $\{g > \lambda\}$.

**Standard rescaling.** In order to put this filtered complex in the same footing as geometric complexes, which are usually indexed by the non-negative real numbers $[0, \infty)$, we apply the following standard rescaling. Consider the isomorphism of posets

$$\left([0,\infty), \leq\right) \to \left((0,1], \geq\right)$$
$$r \mapsto \exp(-r),$$

with inverse given by $\lambda \mapsto -\log(\lambda)$.

Then, given a fuzzy cover $g : X \to \Gamma^{k-1}$, we obtain a filtered complex indexed by the non-negative real numbers by considering $\{\mathsf{Ner}_{\exp(-r)}(g)\}_{r \geq 0}$.

## E. Cover learning as a coarse-graining method

Cover learning can be interpreted as a coarse-graining method, in the sense that it provides a coarse representation of geometric data. This connects our work with methods other than those in Section 2, and here we comment on this.

In (Brugnone et al., 2019) and (Huguet et al., 2023), coarsening is done by simplifying the global structure of the point cloud; for instance, by emphasizing cluster structure. The main difference with cover learning is that cover learning produces a graph (or simplicial complex) which encodes the global structure of the point cloud, and which does not have the data points as vertices (but rather groups of data points).

(Pascucci et al., 2007) uses a technique based on the Reeb graph, which is also the main motivation for Mapper, but unlike Mapper it operates on a simplicial complex (or mesh). The main difference with general cover learning is that, since they approximate a Reeb graph, they produce a one-dimensional simplicial complex, and thus, like Mapper (and as explained in Section 2 and Appendix F) it cannot be used to do higher dimensional topological inference.

In (Wolf et al., 2019), the authors present an end-to-end method for single-cell RNA-seq; the relevant part of their method is described in their section "Graph partitioning and abstraction" (pp. 7), where they explain how they coarsen an initial knn graph to a smaller graph; this is done by computing a clustering of the original graph, and then adding edges between

clusters using a measure of connectivity between different clusters. For visualization purposes, their output serves very similar purposes as ours (e.g., Figure 5). Since it is not a goal of theirs, their method is not suitable for higher dimensional topological inference.

## F. Unsuitability of 1D Mapper for higher-dimensional topological inference

We give two explanations for this:

- The initial cover $\{I_i\}_{i=1}^k$ of $\mathbb{R}$ used by 1D Mapper typically does not have triple-intersections (only pairwise intersecting intervals are used). Hence, the pullback cover $\{f^{-1}(I_i)\}_{i \in I}$ does also not have triple-intersections. The cover used by 1D Mapper is a refinement of $\{f^{-1}(I_i)\}_{i \in I}$, and a refinement of a cover with no triple-intersections could in principle have triple-intersections. However, the refinement is given by partitioning each cover element $f^{-1}(I_i)$ into disjoint subsets (since it is a clustering), which guarantees that the cover used by 1D Mapper does not have triple-intersections, which in turn implies, by definition, that the nerve does not have $k$-dimensional simplices for any $k \geq 2$.

- There exist consistency results for 1D Mapper, which imply that, under suitable assumption, the nerve of the 1D Mapper cover converges to the Reeb graph (Reeb, 1946) of an underlying manifold $\mathcal{M}$ with respect to a function $\varphi : \mathcal{M} \to \mathbb{R}$, see, e.g., Thm. 7 (Carrière et al., 2018). A Reeb graph is a graph, and hence does not contain higher dimensional information.

What this means is that the nerve of the 1D Mapper cover, by design, only contains 0- and 1-dimensional information, that is, it can detect cluster structure and loops in data, but it cannot be used to, e.g., estimate higher-dimensional homology.

## G. Witness complex and Ball Mapper

Let $X$ be a finite metric space, and let $Y \subseteq X$ be a subset. The witness complex with parameter $v = 0$, introduced in (De Silva & Carlsson, 2004), is a filtered simplicial complex $\{W_\infty(Y, X)_\varepsilon\}_{\varepsilon \geq 0}$, with vertices the elements of $Y$, and a simplex on vertices $y_1, \ldots, y_k$ at scale $\varepsilon$ if there exists $x \in X$ such that $d_X(y_i, x) \leq \varepsilon$ for every $i \in \{1, \ldots, k\}$.

**Proposition G.1.** *Let $X$ be a finite metric space, let $\varepsilon \geq 0$, and let $Y \subseteq X$ be an $\varepsilon$-net of $X$. Then the nerve of the Ball Mapper cover constructed using the $\varepsilon$-net $Y$ is equal to $W_\infty(Y, X)_\varepsilon$.*

*Proof.* The two simplicial complexes have the same set of vertices. By definition, there is a simplex in $W_\infty(Y, X)_\varepsilon$ on vertices $y_1, \ldots, y_k \in Y$ if and only if there exists $x \in X$ such that $d_X(y_i, x) \leq \varepsilon$ for every $i \in \{1, \ldots, k\}$, which is equivalent to $\bigcap_{i=1}^k B(y, \varepsilon) \neq \emptyset$, and this is equivalent to there being a simplex on the same vertices in the nerve of the Ball Mapper cover. $\square$

Thus, the filtration output by the multiscale Ball Mapper algorithm of (Dłotko, 2019a) consists of a finite set of filtration values of the Witness complex with parameter $v = 0$.

## H. More on loss estimation

### H.1. Loss normalization

Let $W = \sum_{(x,y) \in G} w(x, y)$ be the total edge weight, and define

$$\eta_M := \frac{1}{k \cdot |X|^2}$$
$$\eta_G := \frac{1}{k \cdot W^2}$$
$$\eta_T := \frac{1}{k \cdot |X|}$$
$$\eta_R := \frac{1}{k \cdot W}.$$

## H.2. Regularization bounds geometry loss

In our experiments, the geometry loss $\widehat{\mathsf{G}}$ usually plays little role, in the sense that results are very similar with or without it. We believe that this is because the regularization $\widehat{\mathsf{R}}$ is sufficient to ensure geometric regularity; here we give weak evidence of this, by showing that $\widehat{\mathsf{G}} \leq \widehat{\mathsf{R}}$.

Let $X$ be a finite set, let $G$ be a weighted graph with $X$ as vertex set, let $E$ be the set of edges of $G$, and let $W = \sum_{e \in E} w(e)$ be the total edge weight. If $F : E \to \mathbb{R}$, then

$$\left( \frac{1}{W} \cdot \sum_{e \in E} w(e) \cdot F(e) \right)^2 \leq \frac{1}{W} \cdot \sum_{e \in E} w(e) \cdot F(e)^2 \,, \tag{2}$$

by Jensen's inequality.

If $g : X \to \Gamma^{k-1}$ is a fuzzy cover, letting $F(x, y) = |g_i(x) - g_i(y)|$ and summing over $1 \leq i \leq k$ gives $\widehat{\mathsf{G}}(g) \leq \widehat{\mathsf{R}}(g)$, by Equation (2).

# I. More on implementation and examples

## I.1. Software

### I.1.1. IMPLEMENTATION

Our implementation of ShapeDiscover (Scoccola & Lim, 2025) is in PyTorch (Paszke et al., 2019), and we rely on Numpy (Harris et al., 2020), Scipy (Virtanen et al., 2020), Numba (Lam et al., 2015), Scikit-learn (Pedregosa et al., 2011), and Gudhi (The GUDHI Project, 2015).

Visualizations are done with Matplotlib (Hunter, 2007), Networkx (Hagberg et al., 2008), and Pyvis (Perrone et al., 2020).

### I.1.2. EXPERIMENTS

We compute Vietoris–Rips filtrations using the Python frontend (Tralie et al., 2018) for Ripser (Bauer, 2021). For edge collapse and sparse Vietoris–Rips we use the implementations in (The GUDHI Project, 2015), and for discrete Morse theory on filtered complexes we use Perseus (Nanda, 2012). We use the KeplerMapper implementation of the Mapper algorithm (van Veen et al., 2019).

All experiments were run on a MacBook Pro with Apple M1 Pro processor and 8GB of RAM.

## I.2. Datasets

**Synthetic data.** These consist of 2000 point uniform samples of the unit 2- and 3-spheres. For the topological inference example, the Betti numbers to be recovered are $\{1, 0, 1\}$, and $\{1, 0, 0, 1\}$, respectively.

**Human in 3D.** This consists of a 9508 point sample of the surface of a human figure in three dimensions (see Figure 7). Topologically, it is thus equivalent to a 2-sphere, and in the topological inference example the Betti numbers to be recovered are $\{1, 0, 1\}$.

**Video dynamical system.** The dataset is from (Lederman & Talmon, 2018), and consists of a video of two figurines rotating at different speeds. When interpreting it as a dynamical system, the state space consists of two independent circular coordinates, and thus of a topological 2-torus; this is confirmed in (Scoccola et al., 2023) using persistent homology. We use the same preprocessing as (Scoccola et al., 2023), which results in a point cloud with 2022 points in 10 dimensions. For the topological inference example, the Betti numbers to be recovered are $\{1, 2, 1\}$.

**Population activity in grid cells data.** The dataset is from (Gardner et al., 2022), and we follow their preprocessing steps to obtain a point cloud with 10000 points in 6 dimensions. As is confirmed in (Gardner et al., 2022) using UMAP and persistent homology, the point cloud is concentrated on a topological 2-torus, so the Betti numbers to be recovered are $\{1, 2, 1\}$.

*Table 3.* Correspondence between labels in Figures 5 and 16 and the cell types of (Packer et al., 2019).

| Label | 1 | 2 | 3 | 4 | 5 | 6 | 7 | 8 | 9 | 10 | 11 | 12 | 13 | 14 | 15 | 16 | 0 |
|---|---|---|---|---|---|---|---|---|---|---|---|---|---|---|---|---|---|
| Cell type | ADF | ADL | ADF_AWB | AFD | ASE | ASER | ASE_par | ASG | ASH | ASI | ASJ | ASK | AUA | AWA | AWB | AWC | all other |

**Octopus in 3D.** This consists of a 7812 point sample of the surface of an octopus figure in three dimensions (see Figure 7).

**Small handwritten digits datset.** This is the dataset of (Alpaydin & Kaynak, 1998), consisting of 5620 handwritten digits encoded as vectors in 64 dimensions.

**MNIST.** This is the classical dataset of (Deng, 2012). We use the training data, which consists of 60000 handwritten digits encoded as vectors in 784 dimensions.

**C. elegans data.** This is the dataset of (Packer et al., 2019), which we preprocess using Monocle 3 as in (Monocle3). This results in a point cloud with 6188 points in 50 dimensions. There are 28 cell types. To help visualization, we keep 16 cell types (those whose flare structure is clearly identified by UMAP's 2D projection; see Figure 16), which we label from 1 to 16, and label the rest of the cell types by 0; see Table 3.

**Other comments.** In the topological inference experiments, we scale datasets so that the sparsification parameter for sparse Vietoris–Rips can be fixed.

### I.3. Details on efficient topological representation experiment

We compare ShapeDiscover to the following approaches to topological inference:

- (S + Rips) Furthest point subsample + Vietoris–Rips.

- (S + Rips) Furthest point subsample + Vietoris–Rips + discrete Morse theory on filtrations (Mischaikow & Nanda, 2013).

- (S + Rips + edg.coll.) Furthest point subsample + Vietoris–Rips + edge collapse (Boissonnat & Pritam, 2020).

- (S + sparse Rips) Furthest point subsample + sparse Vietoris–Rips (Sheehy, 2013).

- (S + Alpha) Furthest point subsample + Alpha complex (Boissonnat et al., 2018).

- (Witness) Witness complex (De Silva & Carlsson, 2004). Recall from Appendix G that, when $v = 0$, the Witness complex is essentially equivalent to multiscale Ball Mapper (Dłotko, 2019a) (MS BMapper).

Furthest point subsampling refers to the strategy for building a subsample of a point cloud (or finite metric space) given by choosing an initial point and iteratively choosing the point that maximizes the (minimum) distance to the points already chosen.

OOM means out of memory. The implementation of discrete Morse theory on Vietoris–Rips complexes we use (Nanda, 2012) runs out of memory on larger datasets; this implementation also does not report number of vertices of the final simplicial complex, hence the "-" in the table. Alpha complexes do not scale well with ambient dimension, which is reflected in the fact that it runs out of memory in the dynamical system dataset, which has ambient dimension 10.

### I.4. Homology recovery quotient

This is a simplified version of the quotient used to quantify topological recovery in (De Silva & Carlsson, 2004). Given a filtered simplicial complex $\mathbb{K} = \{K_r\}_{r \in \mathbb{R}}$, we let $a \in \mathbb{R}$ (resp. $b \in \mathbb{R}$) the smallest (resp. largest), finite, left (resp. right) endpoint of bars in the barcode of $\mathbb{K}$. Then, given target Betti numbers $\{\beta_0, \ldots, \beta_n\}$, the homology recovery quotient of $\mathbb{K}$ is defined as:

$$\frac{|\{r \in \mathbb{R} : \dim \mathsf{H}_i(K_r) = \beta_i, \forall 0 \leq i \leq n\}|}{b - a},$$

where in the numerator we are taking the Lebesgue measure.

### I.5. Why not use Mapper in the topological inference experiment

As explained in Section 2 and Appendix F, 1D Mapper does not contain higher-dimensional information. Higher-dimensional extensions of Mapper exist, but the parameters are difficult to tune, and require higher-dimensional filter functions. Mapper is also static, that is, it produces a single simplicial complex. In order to produce a sequence of simplicial complexes, and thus a barcode that can be compared with standard approaches to topological inference, one would need to use multiscale Mapper. But to the best of our knowledge there is no implementation of a multiscale, higher-dimensional Mapper. We expect the parameters of such a procedure to be very hard to tune; this could be alleviated by using the framework of (Oulhaj et al., 2024), to produce a differentiable, multiscale, higher-dimensional Mapper.

### I.6. Details on toroidal topology in neuroscience experiment

**Result.** We run ShapeDiscover on the data of (Gardner et al., 2022) (see Appendix I.2 for details), with n_cov = 15; resulting in a filtered complex with 15 vertices. The barcode thus obtained is in Figure 6, and the homology recovery quotient is 0.2, which is remarkable given that we only used a cover with 15 elements.

**Comparison to Vietoris–Rips.** We also run furthest point subsaple + Vietoris–Rips on the dataset. In our experience, it is not possible to uncover the toroidal topology by directly using Vietoris–Rips: we subsampled 5000 points and computed the Vietoris–Rips barcode, which resulted in a homology recovery quotient of 0, that is, in a filtration which at no point has toroidal topology. This agrees with the methodology of (Gardner et al., 2022), where the toroidal topology was uncovered with persistent homology only after a non-linear dimensionality reduction step using UMAP.

**Comparison to other toroidal topology dataset.** In our other experiment on a toroidal dataset (the dynamical system data of Section 5.1), ShapeDiscover required a cover with 52 elements to clearly recover the torus. We believe this is because, in that dataset, the aspect ratio of the torus is highly skewed, since one of the figurines is significantly larger than the other one (see Fig 2 (Lederman & Talmon, 2018)), which implies that, in the state space of the dynamical system, one of the two circular coordinates is metrically larger than the other one.

### I.7. Plotting the nerve of a cover

We plot the nerve of a cover $\{U_i\}_{i \in I}$ with the following conventions:

- The size of a vertex corresponding to $U_i$ is proportional to $\log(|U_i| + 1)$.

- The thickness of an edge corresponding to $U_i \cap U_j$ is proportional to $\log(|U_i \cap U_j| + 1)$.

- The length of an edge corresponding to $U_i \cap U_j$ is proportional to $1/|U_i \cap U_j|$.

- When the data has labels, the vertex corresponding to $U_i$ is plotted as a pie chart, which shows the proportion of elements of each class in $U_i$ (note that labels are not used to produce the cover). This is a standard practice in for Mapper graphs, see, e.g., (Chalapathi et al., 2021; Zhao et al., 2020).

### I.8. Differentiable Mapper on visualization experiments

We experimented with the implementation of Differentiable Mapper (Oulhaj et al., 2024) provided in (Oulhaj, 2024) on the digits and C. elegans data, but we were unable to obtain meaningful results. As parametric family of filter functions, we used a linear projection, as used in (Oulhaj et al., 2024). Lack of success could be due to the difficulty of learning a filter function which properly reflects the structure of the datasets, and it remains a possibility that a more sophisticated parametric family of filter functions (as suggested in the conclusions of (Oulhaj et al., 2024)) with finer parameter tuning will lead to good results.

### I.9. Other analyses

See Table 5 for an ablation study, Figure 18 for a parameter sensitivity analysis, and Figure 19 for a noise sensitivity analysis.

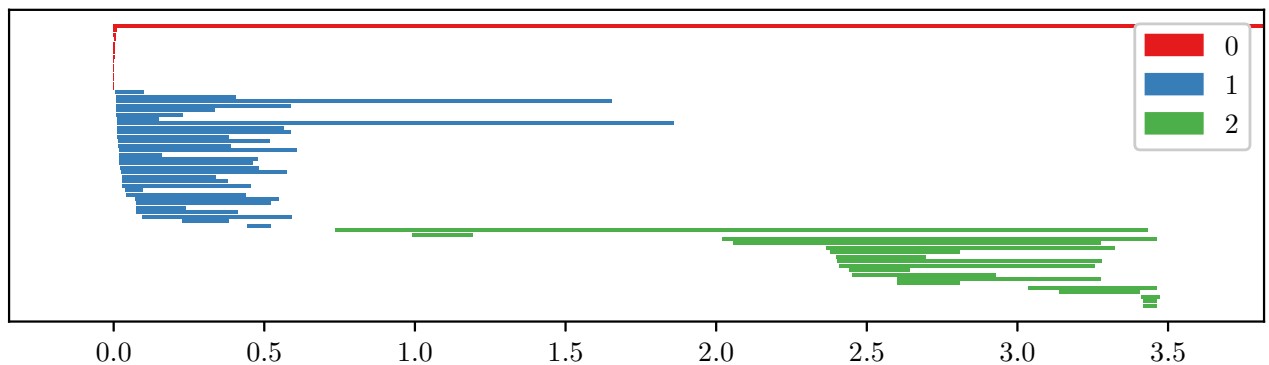

*Figure 6.* The barcode of the filtered simplicial complex obtained by running ShapeDiscover with a cover with 15 elements on the dataset of (Gardner et al., 2022). The toroidal topology is apparent between scales 1.25 and 2, and the homology recovery quotient is 0.2. Compare with Fig 1 (Gardner et al., 2022).

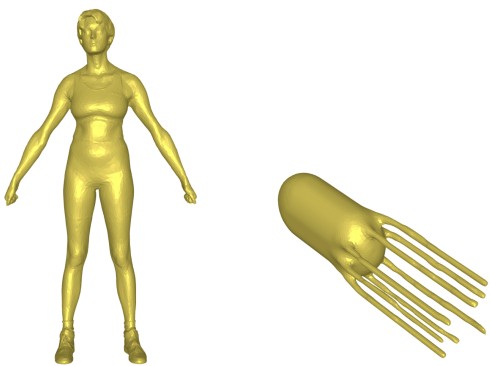

*Figure 7.* Human and octopus datasets projected to two dimensions. Figures taken from (Oulhaj et al., 2024).

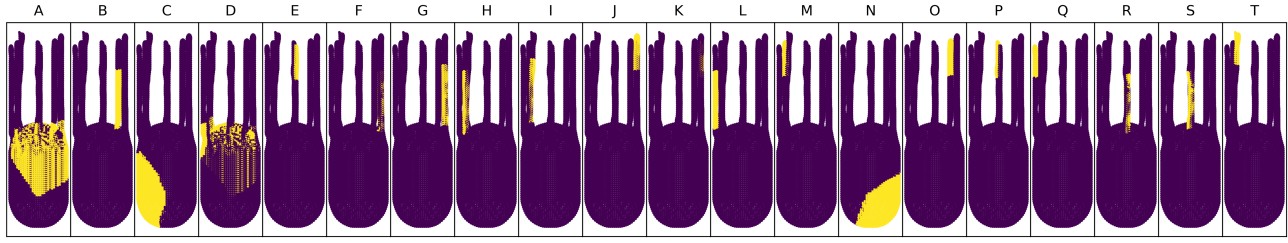

*Figure 8.* Cover of the octopus data obtained with ShapeDiscover with default parameters and n_cov = 20.

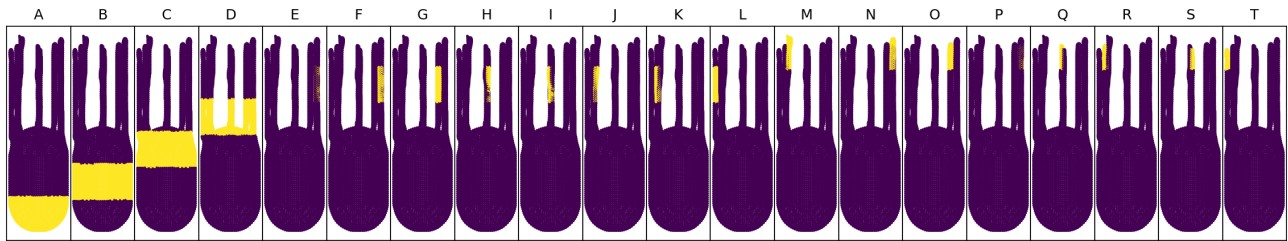

*Figure 9.* Cover of the octopus data obtained with Mapper.

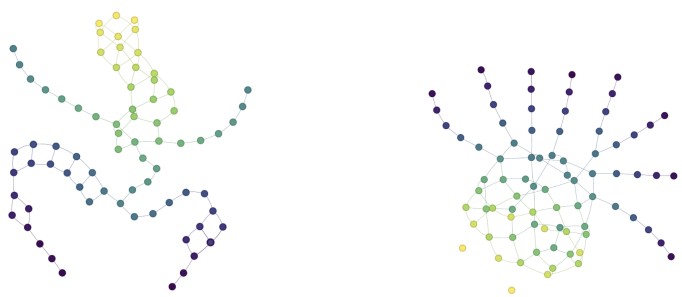

*Figure 10.* Differentiable Mapper on human and octopus data. Figures taken from (Oulhaj et al., 2024).

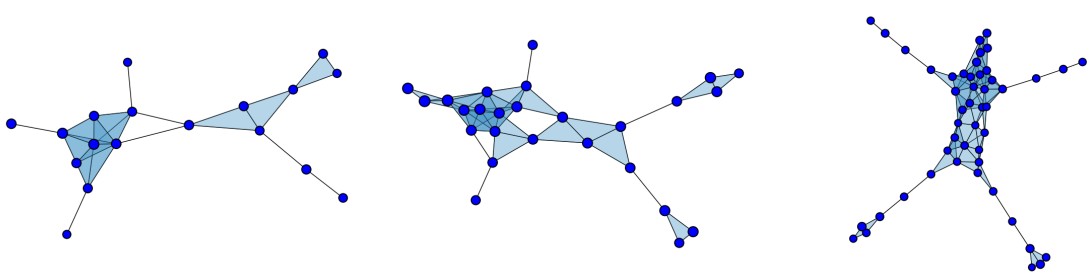

*Figure 11.* Ball Mapper on human data with parameter (left-to-right): $\varepsilon = 0.25$, $\varepsilon = 0.2$, $\varepsilon = 0.15$. The cover sizes are 18, 26, 48, respectively. We see that the $\varepsilon$ parameter depends significantly on the data, and is hard to tune. The underlying topology (that of a 2-sphere) is also hard to recover: none of the simplicial complexes have the underlying topology, which agrees with the findings in Table 1 (row "Witness $(v = 0)$ / MS BMapper").

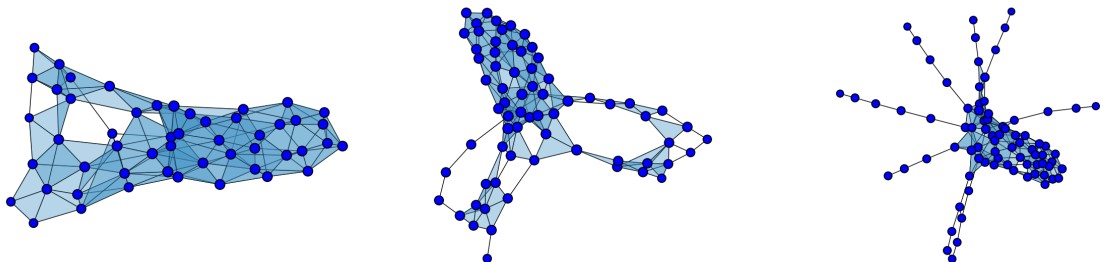

*Figure 12.* Ball Mapper on octopus data with parameter (left-to-right): $\varepsilon = 0.2$, $\varepsilon = 0.175$, $\varepsilon = 0.15$. The cover sizes are 51, 68, 80, respectively. The same conclusions as in Figure 11 hold.

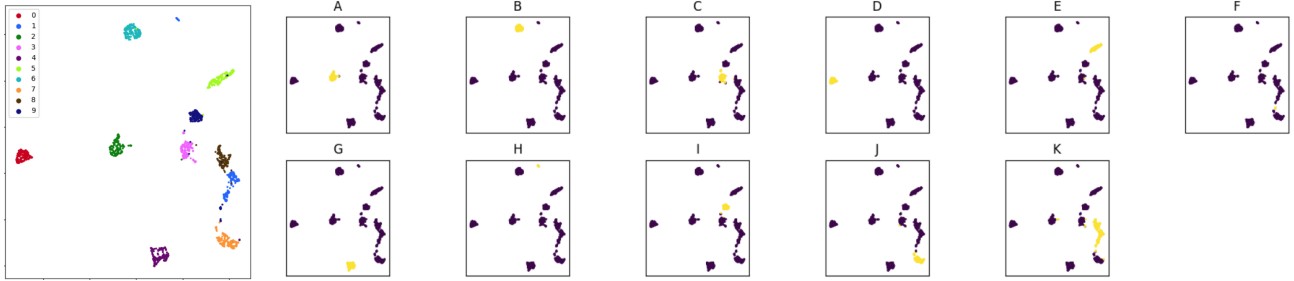

*Figure 13. Left.* 2D projection of the digits data obtained with UMAP. *Right.* Cover of the digits data obtained with ShapeDiscover, plotted on the 2D projection of the data obtained with UMAP.

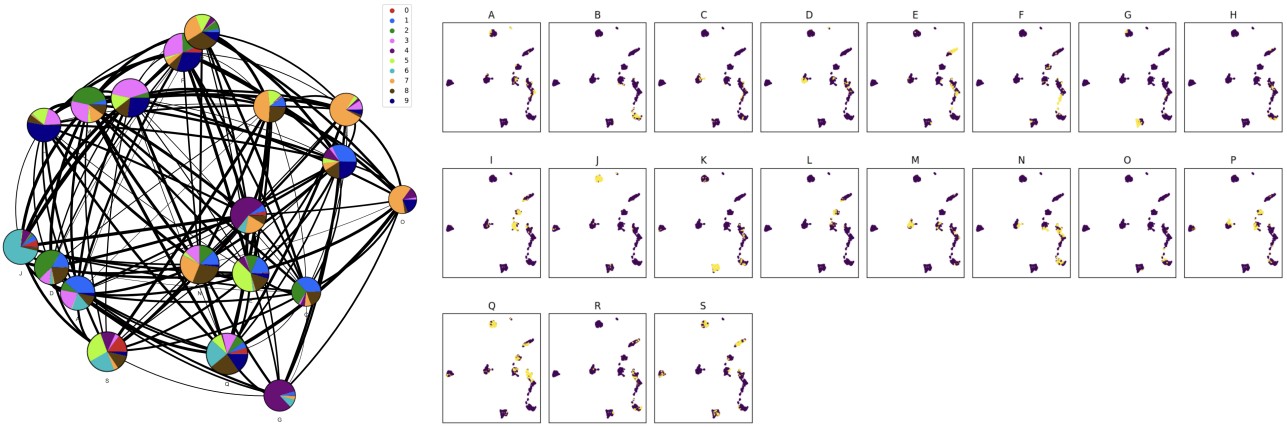

*Figure 14.* Ball Mapper on digits data with $\varepsilon = 45$, which results in a cover of size 19. The (1-skeleton) of the nerve of the cover (left), and the cover plotted on the 2D projection of the data obtained with UMAP (left); see Figure 4. We see that different digit are mostly mixed up, and there cover elements overlap with many others, presumably due to the curse of dimensionality.

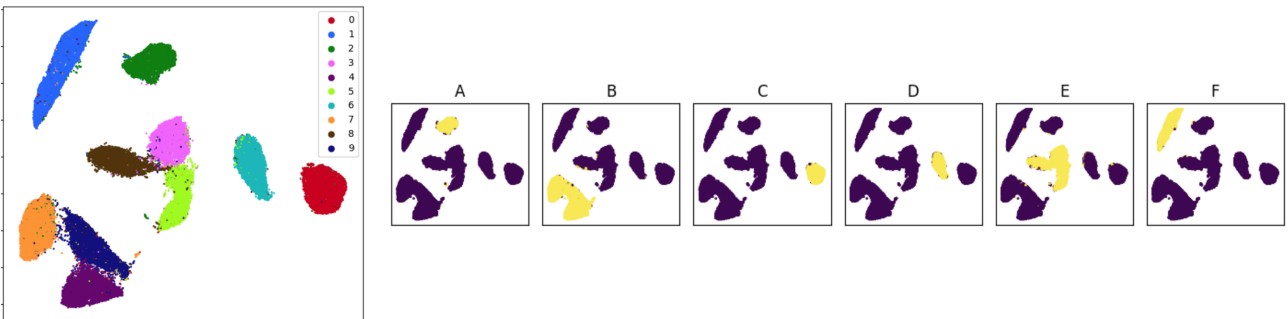

*Figure 15. Left.* 2D projection of MNIST obtained with UMAP. *Right.* Cover of MNIST obtained with ShapeDiscover, plotted on the 2D projection of the data obtained with UMAP.

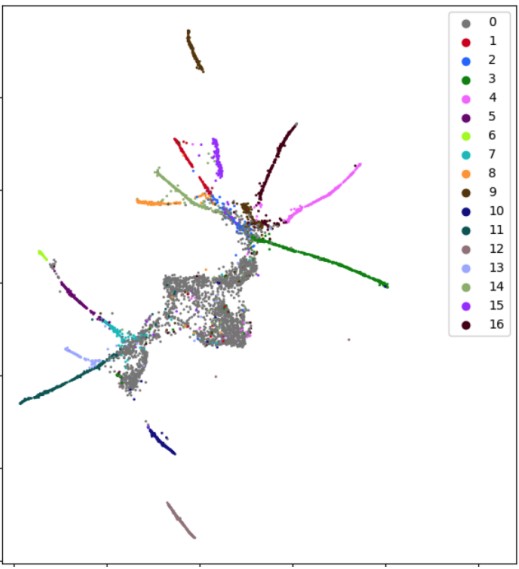

*Figure 16.* C. elegans data projected with UMAP. Labels are described in Table 3.

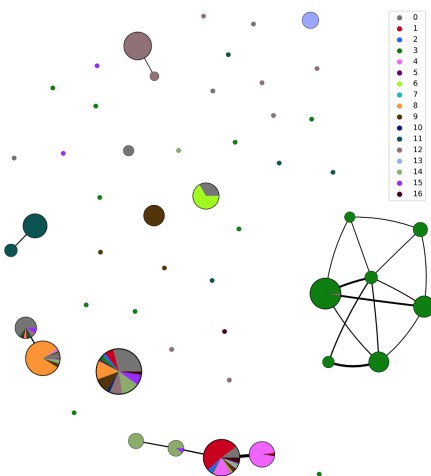

*Figure 17.* Ball Mapper on C. elegans data with $\varepsilon = 70$, which results in a cover of size 51. There is a large, independent, cover element with contains most points. In order to break it apart, we must make $\varepsilon$ smaller, but this results in more and more tiny, independent cover elements. The flare structure is not recovered.

*Table 4.* Times (in seconds) taken to run ShapeDiscover on the datasets of the examples. Columns correspond to datasets together with the `n_cov` parameter of Table 1. Rows correspond to ShapeDiscover's subroutines. We see that the bulk of the time is taken by the optimization step, and that the topology loss accounts for a large proportion of this.

| ShapeDiscover subroutine | 2-sph 5 | 3-sph 19 | human 19 | dynam 52 | grid cell 15 | octopus 20 | digits 15 | MNIST 15 | C elegans 100 |
|---|---|---|---|---|---|---|---|---|---|
| Preprocessing | 0.05 | 0.04 | 0.13 | 0.05 | 0.17 | 0.12 | 0.05 | 4.37 | 0.27 |
| Clustering | 0.04 | 0.04 | 0.32 | 0.10 | 0.31 | 0.55 | 0.04 | 2.96 | 0.94 |
| Optimization | 1.74 | 4.33 | 3.95 | 2.38 | 4.31 | 2.56 | 2.13 | 206.04 | 77.24 |
| Opt. w/o topology loss | 1.05 | 1.55 | 1.18 | 0.66 | 1.51 | 0.82 | 0.61 | 50.03 | 19.83 |

*Table 5.* Rows correspond to datasets together with the `n_cov` parameter of Table 1. Columns correspond to components of the pipeline that are turned off. Entries are the output homology recovery quotient. We observe that the measure and regularization losses are essential. The initialization using clustering is also essential, at least for real datasets. The geometry loss has no substantial effect, at least for these datasets. The topology loss has an effect, but only when the initialization is random.

| Dataset | full pipeline | no meas. loss | no geom. loss | no top. loss | no reg. loss | random init. | random init. + no top. loss |
|---|---|---|---|---|---|---|---|
| 2-sph 5 | 0.74 | 0 | 0.72 | 0.74 | 0 | 0.59(0.1) | 0.1(0.03) |
| 3-sph 19 | 0.51 | 0 | 0.52 | 0.51 | 0 | 0.16(0.03) | 0.03(0.01) |
| human 19 | 0.7 | 0 | 0.71 | 0.69 | 0 | 0 | 0 |
| dynam 52 | 0.38 | 0 | 0.41 | 0.38 | 0 | 0 | 0 |

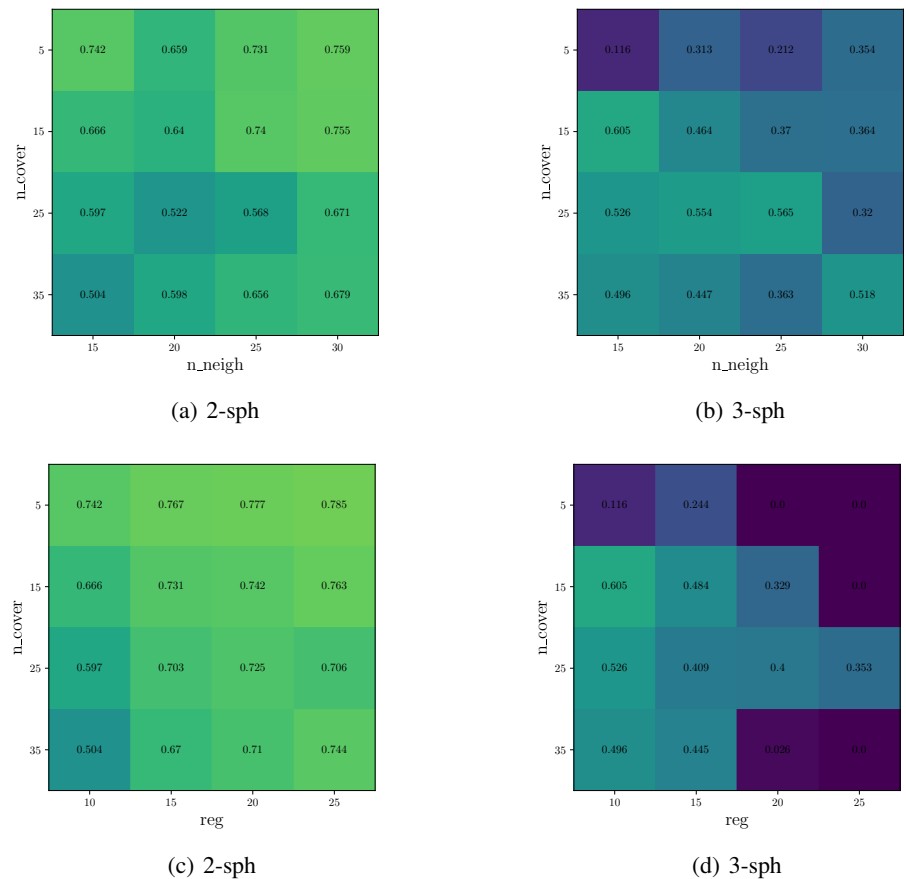

(a) 2-sph             (b) 3-sph

(c) 2-sph             (d) 3-sph

*Figure 18.* We run ShapeDiscover on the 2- and 3-sphere datasets, with parameters as in Table 1, except for the ones being displayed on the axes, and compute homology recovery quotient (entries).

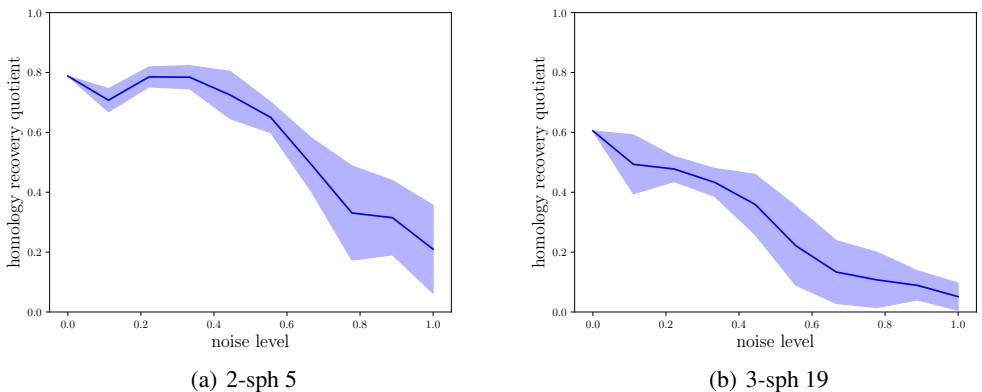

(a) 2-sph 5             (b) 3-sph 19

*Figure 19.* We run ShapeDiscover on the 2- and 3-sphere datasets, with parameters as in Table 1, and compute homology recovery quotient. Data is normalized to exactly fit in a unit hypercube. Noise is additive, meaning that each coordinate of each data point is perturbed by a uniform random number in $[0, 1]$. We display mean and standard deviation over 10 runs for each level of noise.

