# OpenReview forum: "Cover learning for large-scale topology representation"
_ICML.cc/2025/Conference — ICML 2025 poster_

### Official Review · Reviewer_kSTd · 2025-03-10

**Overall Recommendation:** 3

**Summary:**

This paper introduces "cover learning," an innovative unsupervised learning method designed to represent the large-scale topological structure of geometric datasets. It extends and addresses limitations present in traditional Topological Data Analysis (TDA) methods, specifically those relying on geometric complexes and Mapper graphs. The authors identify the fundamental optimization challenge of learning topologically-faithful covers and provide theoretical grounding for optimizing such covers using fuzzy covers, which facilitate gradient-based optimization. The paper proposes ShapeDiscover, a practical algorithm utilizing fuzzy cover optimization, which outperforms standard TDA methods in both quantitative topological inference and qualitative topology visualization.

**Claims And Evidence:**

Yes.

**Essential References Not Discussed:**

N/A.

**Experimental Designs Or Analyses:**

**W1.** It seems that the experiments mainly use default parameters and vary only two parameters (maximum cover size and threshold $\lambda$). A systematic sensitivity analysis would help in clarifying how sensitive the performance of ShapeDiscover is to variations in parameters, providing more guidance on parameter tuning.

**W2.** While diverse datasets are considered, there is no explicit analysis of how ShapeDiscover performs in the presence of noise, which is a critical consideration for practical data analysis. Please include experiments (or theoretical results) with varying levels of artificial noise and assess the robustness and stability of the inferred topological structures.

**W3.** A valuable avenue not explored is the integration of ShapeDiscover-generated topological representations with downstream machine learning tasks, especially deep learning. For example, investigating how ShapeDiscover’s topological representations could enhance the performance of neural networks on classification tasks (e.g., MNIST) would substantially broaden the practical relevance of this work.

**Methods And Evaluation Criteria:**

Yes.

**Other Comments Or Suggestions:**

N/A.

**Other Strengths And Weaknesses:**

Strengths:

**S1.**	The paper introduces a robust theoretical framework linking cover learning to fuzzy cover optimization, grounded firmly in geometry, topology, and optimization theory.

**S2.** ShapeDiscover demonstrates superior performance in recovering accurate topological structures with fewer simplices compared to traditional methods like Vietoris–Rips and Mapper.

**S3.** ShapeDiscover is capable of handling larger datasets more efficiently compared to existing methods, making it practical for real-world applications in large-scale topology visualization.

Weakness:

**W4.** Despite improvements, the topological persistence optimization remains computationally intensive, potentially limiting scalability for exceptionally large datasets.

**Questions For Authors:**

**Q1.** The algorithm primarily utilizes 0-dimensional homology for computational efficiency, how well does it generalize to higher-dimensional homology inference?

**Relation To Broader Scientific Literature:**

The paper situates itself within the broader scientific domain of TDA, explicitly addressing limitations of established methods such as geometric complexes and Mapper graphs. It connects to foundational concepts in computational geometry and algebraic topology, especially the nerve theorem and persistence theory (Edelsbrunner & Harer, 2022; Oudot, 2015). Moreover, by employing fuzzy covers and optimization frameworks common in modern machine learning, the paper bridges classical topology with contemporary computational methods, facilitating broader applicability.

**Theoretical Claims:**

To the best of my knowledge, the proofs are correct.

---

> ### Author Rebuttal · Authors · 2025-03-27
>
> Thank you for your thoughtful feedback!
>
> - W1: This will be addressed in the ablation study/sensitivity analysis we will perform.
> Please see "Main comment A" in the response to reviewer mPj7 for what we plan to include, as well as preliminary findings.
>
> - W2:
> We will include this.
> In our experience so far, the algorithm is robust to noise for both visualization and topological inference purposes, when noise is additive, or not too many outliers are present.
> We will assess this methodically, and see how much and what type of noise is required to break the algorithm.
>
> - W3: We are actively working on this; please see "Main comment C", in the response to reviewer 5NTw.
>
> - W4:
> It is true that topological persistence optimization is computationally intensive, and hasn't been tested on exceptionally large data.
> We want to point out that 0-dimensional persistence optimization is feasible on medium size data.
> In the MNIST experiment, we are running the topology loss on the full dataset, and on 15 cover elements (ie 15 independent times), for ~250 iterations, which effectively means that we run the topology loss 250 * 15 = 3750 times on a dataset with 60,000 points, and this took ~150 seconds.
> We expect scalability to larger data to not pose a big problem, since the (0-dimensional) topological loss has complexity O(n log n) (n being vertices + edges in the knn graph which is in O(nk)).
> There is an easy way to reduce computation time significantly: Run the topology loss stochastically, say every ~10 iterations, which would cut down topology optimization to 15 seconds, in the above example
> (since we use topological optimization with big steps, the gradient of the topological loss is not sparse, so running the loss even a few times does achieve the desired effect).
> Although we have experimented with this, we do not mention it the paper since we want to focus on the main message, and leave further tuning, optimization, and other approaches to cover learning for future work.
> Having said this, higher-dimensional persistence optimization is, at this point in time, significantly less efficient, but we believe there is hope; please see our response to Q1.
>
> - Q1:
> Experiments A and B have examples in which the method uncovers the underlying 0-, 1-, 2- and 3-dimensional homology with many fewer simplicies than the main (to our knowledge) approaches to homological inference, even though our topological loss only ensures connectivity (as it only uses 0-dimensional homology).
> So our method is good at higher-dimensional homological inference even if the topological loss is only 0-dimensional; we believe that this is because small measure, plus regular geometry, plus topological connectivity, in practice, is forcing the cover elements to be contractible, but at this point this is a conjecture.
> Using higher dimensional homology in the topological loss would be computationally more expensive; we expect that, with several standard computational shortcuts (mostly subsampling and topological optimization with big steps), 1-dimensional homology could be added to the loss and have the algorithm run in the order of minutes on datasets of ~10,000 points.
> Moreover, we believe there is hope for faster approaches:
> As we mention in the conclusions, topological regularization only seeks to simplify topology, and thus one does not need to compute, e.g., a full persistence diagram, it is enough to find a persistent feature and to produce a gradient that will make it less persistent;
> we hope that probabilistic algorithms, sparse spectral methods, or ideas like the ones in (Chen, Kerber, Computational Geometry, 2013) could be used for this.

---

> > ### Comment · Reviewer_kSTd · 2025-04-04
> >
> > Thank you for your responses.
> >
> > After carefully reviewing the manuscript and all accompanying reviews and rebuttals, I have decided to maintain my score and recommend a weak acceptance of the paper.

---

### Official Review · Reviewer_5NTw · 2025-03-14

**Overall Recommendation:** 4

**Summary:**

This paper proposes a novel algorithm for learning subset cover of a dataset with respect to its geometric and topological properties. The authors develop a gradient optimization procedure for learning the fuzzy cover of a dataset with required properties; fuzzy cover induces a simplicial filtration (by grade of membership), and by thresholding it at certain pre-defined level, they obtain the desired (crisp) cover. Experiments on different datasets show that the proposed approach yields better covers than other existing algorithms.

## update after rebuttal

I think that with the proposed changes this paper will be an interesting contribution to the research area. I will keep my original score (of 4).

**Claims And Evidence:**

Main claims of the paper are supported with enough evidence.

**Essential References Not Discussed:**

None, as far as I know.

**Experimental Designs Or Analyses:**

Seems reasonable.

**Methods And Evaluation Criteria:**

The proposed algorithm is adequate for the problem of cover learning.

**Other Comments Or Suggestions:**

Can you provide some information on time and memory complexity of the proposed algorithm (theoretical or empirical estimations)?

**Other Strengths And Weaknesses:**

Strengths:
1. Solid theoretical part.
2. Large number of experiments of diverse collection of datasets.
3. Extensive background information provided in Appendices introduces reader to mathematical concepts that are used in the paper.

Weaknesses:
1. Some uncertainty about potential practical applications of the proposed method.

**Questions For Authors:**

1. Are there any other applicable quality metrics, aside from number of vertices and simplices reported in Tables 1 and 2?
2. Can the cover constructed by the optimization procedure be influenced by initialization? If so, can you provide the mean and variance for values in Tables 1 and 2?

**Relation To Broader Scientific Literature:**

The task of cover learning is relatively unexplored at Topological Data Analyses, but something similar was previously covered in other works; however, the problem setting and the approach proposed here are highly novel. All related works are cited in the paper.

**Theoretical Claims:**

All mathematics statements are given with valid proofs.

---

> ### Author Rebuttal · Authors · 2025-03-27
>
> Thank you for your thoughtful feedback!
>
> - About time and memory complexity:
> On Table 4 in the Appendix, we report the times it took to run our experiments.
> Separately to this, we will include a complexity analysis in the paper.
>
> - About practical applications: please see "Main comment C", at the end of this response.
>
> - About other applicable quality metrics for Tables 1 and 2:
> Our goal was to do topological inference with small simplicial complexes (much smaller than SOTA), and we do not know of other standard ways of quantifying simplicial complex size besides number of simplices.
> Please let us know if there are other measures we can compute.
>
> - About dependence on initialization:
> When our method is initialized with a good clustering, results are stable across runs, and, for experiment A, there is indeed no variance.
> When the method is initialized with a random fuzzy cover, the results can vary across runs.
> This will be addressed in the ablation study/sensitivity analysis we will perform.
> Please see "Main comment A" in the response to reviewer mPj7 for what we plan to include, as well as preliminary findings.
>
> **Main comment C: Applications and downstream tasks**
>
> We start by explaining the rationale behind our choice of applications.
> A main selling point of cover learning is that it improves upon the two main TDA methodologies with a single approach, by effectively addressing their main shortcomings: the sometimes prohibitive size of geometric complexes, and the difficulty in tuning and the lack of higher dimensional information in Mapper graphs.
> The two applications in the paper (sections 5.1 and 5.2) were chosen to demonstrate this.
>
> The effectiveness in downstream tasks (eg shape classification) of the TDA methodologies we are improving upon has been established in the literature, so we decided to limit the scope of the paper to emphasize the theory we develop (Section 3), and the computational methods to make optimization feasible (Section 4).
> Let us also mention that, besides machine learning, topological inference has had significant scientific applications such as (Gardner et al., Nature 2022) and (Benjamin et al., Nature 2024), and that Experiment B shows that our method (re)discovers the main finding of the former paper with ease.
>
> In addition to the above, here are further applications we are actively exploring, and will be happy to mention in the paper:
> - Point cloud vectorization:
> Typically, point clouds ought to be vectorized in a permutation-invariant way.
> One way to do this is to construct a graph on the point cloud (eg knn graph) and then apply methods from graph machine learning.
> Large point clouds result in large graphs, from which the global geometry is hard to extract (due to high dimensionality of the learning problem).
> Our method provides much smaller graph/simplicial complex representations of point clouds, which can potentially simplify the learning step (eg by requiring less training data).
> - Local+global dimensionality reduction:
> Popular modern dimensionality reduction algorithms such as tSNE and UMAP operate essentially locally, in the sense that they optimize embedding using interactions between few data points (typically two); with these algorithms, the preservation of global structure happens as a by-product of the preservation of global structure.
> We believe that a local optimization procedure, as above, could be combined with a global one which is tasked with ensuring preservation of global topology, as captured by a small simplicial complex given as the nerve of a cover learned from the data.
> - Simplifying parameter selection in clustering:
> A main difficulty with many standard clustering algorithms (eg k-means) is that the number of final clusters needs to be chosen by the user.
> An l-element cover gives rise to a graph with l vertices, and if the cover is good, the connected component of this graph represent the large scale cluster structure of the data, and give rise to a clustering of the data. So the chosen number of cover elements l is just an upper bound for the number of output clusters, and the final clustering (and the number of clusters) depends on the intrinsic geometry of the data, and not on the arbitrarily chosen number l.
> - Input for simplicial neural networks:
> Cover learning produces a compact simplicial complex on geometric data, and can thus be combined with simplicial neural networks, such as
> (Roddenberry et al, ICML 2021),
> (Chen et al, AAAI, 2022),
> (Maggs et al, ICLR 2024),
> (Gurugubelli, Chepuri, ICLR 2024).

---

> > ### Comment · Reviewer_5NTw · 2025-04-04
> >
> > Thank you for the provided clarification. I have also read other reviews and responses to them. I think that with the proposed changes this paper will be an interesting contribution to the research area. I will keep my original score.

---

### Official Review · Reviewer_Wjc2 · 2025-03-14

**Overall Recommendation:** 3

**Summary:**

The paper aims to generate topologically faithful simplicial complexes for geometric datasets by reducing the problem to cover learning. By formally defining a set of three goals for cover learning, extending to the space of fuzzy covers (“softening” the inclusion of an element in a subset, which allows them to be parametrized by functions over real numbers), and demonstrating a way to estimate these goals as standard loss functions, the authors develop a framework through which these goals can be optimized with standard neural networks. Based on this idea, they implement a cover learning algorithm, ShapeDiscover, which outperforms other models quantitatively by requiring less vertices and simplices to achieve the same homology recovery quotient on synthetic geometric datasets. This model also provides more intuitive topological representations than previous cover learning approaches.

**Claims And Evidence:**

The paper supports all of its mathematical claims with sufficient proofs, either included in the main body or the appendix. Its claims on improvements to visual representation, while less rigorous, are supported by sufficient experimental evidence.

**Essential References Not Discussed:**

All of the ones mentioned in this review:
Huguet et al. 2023
Wolf et al. 2019
Brugone et al. 2019
Pascussi et al
Also topological encoders from Moore et al. 2020

**Experimental Designs Or Analyses:**

I think the proposal here is at the end of the day a kind of coarse graining or sketching of the graph used to compute homology etc. Gie this they should be comparing to other coarse graining methods like diffusion topology from (Huguet et al. 2023 SIAM, brugnone et al. IEEE big data 2019) , also Reeb graph method like Pascussi et al. ACM Trans on graphics, also PAGA from Wolf et al. 2019.

**Methods And Evaluation Criteria:**

The defined goals are reasonable for the problem of cover learning; generating a cover with measure-theoretically small sets, geometrically regular sets, and a homologically faithful nerve are all well-defined and meaningful objectives in cover learning.

**Other Comments Or Suggestions:**

None.

**Other Strengths And Weaknesses:**

Strengths: the paper is very well-structured and intuitive to follow. The initial definition of three main goals in cover learning using words, followed by concise mathematical expressions for each aim, allows for a clear derivation for the paper’s resulting loss function. Representing sets in covers as vertices over a k-dimensional simplex, and representing fuzzy sets as all points contained in this simplex allows for a clean parameterization for these covers using softmax.


Weakness: A lack of ablation studies and comparisons; an interesting study would be to exclude certain components of the loss function and observe impacts on performance. In addition, it would be interesting to see the performance of just the “FuzzyCoverInitialization” using spectral clustering, and see how much further this is optimized after gradient descent.

**Questions For Authors:**

1. From my understanding, the number of vertices (i.e. the number of subsets in the cover) is a fixed parameter; is this correct? For the results in Table 1, does the model have to be completely trained from scratch to test each vertex number 1, 2, … k until the desired homology quotient is achieved?
2. Would this method yield more fruitful results if it used different resolutons of covers in keeping with the topological theme/

**Relation To Broader Scientific Literature:**

The contributions of the paper are highly relevant to topological data analysis, and can be used alongside recent methods which utilize simplicial complexes to draw geometric insights from data, including [Maggs et al. 2024] published in ICLR 2024. To the best of my knowledge, no previous methods have enabled the parameterization of cover learning under traditional machine learning frameworks.

**Theoretical Claims:**

All theoretical claims are supported with correct proofs.

---

> ### Author Rebuttal · Authors · 2025-03-27
>
> Thank you for your thoughtful feedback!
>
> - About missing references: We will include them in the paper and comment on similarities and differences.
> Here are the main ideas we took from the papers (please correct us if we have misinterpreted something):
>     - (Huguet et al. 2023)(Brugnone et al. 2019): Coarsening is done by simplifying the global structure of the point cloud (for example, by emphasizing cluster structure).
>     The main difference is that our method produces a graph (simplicial complex) which encodes the global structure of the point cloud, and which does not have the data points as vertices (but rather groups of points).
>     - (Pascussi et al. 2007): The technique is based on Reeb graph, which is also the main motivation for Mapper, but unlike Mapper it operates on a simplicial complex.
>     The main difference is that, since they approximate a Reeb graph, they produces a one-dimensional simplicial complex, and thus, like Mapper (and as explained in Sec. 2 and App. D) it cannot be used to do higher dimensional topological inference.
>     - (Wolf et al. 2019): This paper presents an end-to-end method for single-cell RNA-seq; the relevant part of their method is described in their section "Graph partitioning and abstraction" (pp. 7),
>     where they explain how they coarsen an initial knn graph to a smaller graph; this is done by computing a clustering of the original graph, and then adding edges between clusters using a measure of connectivity between different clusters.
>     For visualization purposes, their output serves very similar purposes as ours (eg our Fig. 5), and is thus very related.
>     Since it is not a goal of theirs, their method is not suitable for higher dimensional topological inference.
>     - (Moor et al. 2020): This is an autoencoder-based dimensionality reduction algorithm, with the novelty being the usage of a topological loss on top the classical reconstruction loss.
>     Two connections to our work:
>     One is producing a coarsened representation of the data; here the difference is that they output an embedding in low dimensional space, while we output a graph (simplicial complex) with groups of data points as vertices.
>     The second one is the usage of a persistence-based loss; the main difference with our usage is that we use it purely as regularization (we enforce trivial local topology, motivated by the nerve theorem), whereas they use it to directly enforce the topology of the low dimensional representation to be similar to that of the original data.
>     This work will also be mentioned in our section "A.5. Topological persistence optimization".
>
> - About our method being a coarse-graning/sketching method:
> This is a good interpretation.
> We want to emphasize the fact that our coarsening strategy serves two distinct purposes: topological inference (via homology) and visualization.
> In particular, we compute homology directly on the coarse representation (small simplicial complex), as opposed to computing homology of a large initial graph that then is coarsened only for visualization.
> Moreover, our strategy builds on the nerve construction (standard in topology), which lends itself to further theoretical analysis (eg consistency of covering algorithms, addressed in future work).
>
> - About ablation study: This will be implemented.
> Please see "Main comments A and B" in the response to reviewer mPj7 for what we will include, as well as preliminary findings.
>
> - About the number of cover elements parameter:
> The parameter is fixed, and in our experiments we simply retrained the model for each choice each case.
> Two comments:
> First, in practice, the chosen number is just an upper bound for the number of cover elements: gradient descent can converge to solutions in which some cover elements are empty; this a feature as it simplifies the choice of the parameter.
> Second, if an l-element cover is already available, we can use it as initialization to optimize for a (l+n)-element fuzzy cover (eg set the new cover elements uniformly at random and normalize to get a fuzzy cover).
>
> - About different resolutions of covers:
> (We believe we understand what you mean, but please let us know if the following does not address your question.)
> Indeed! And this is what the method does.
> As described in Section 3, paragraph "Fuzzy covers", optimization is done over the space of fuzzy covers, which is just a different name for a persistent/multiresolution/multiscale cover, ie a family of nested covers, in our case indexed by [0,1].
> This is very much in line with persistent topology/TDA, and is what allows us to perform the quantitative experiment A, in which we show that our method compares favorably to all (to our knowledge) TDA approaches to topological inference based on persistent homology.
> The question might be about other applications of the fact that we have a multiscale cover, since, for example, we don't leverage this in the visualization examples.
> This is something we are actively working on.

---

### Official Review · Reviewer_mPj7 · 2025-03-20

**Overall Recommendation:** 3

**Summary:**

This paper focuses on learning a representation of the large-scale structure of geometric datasets, and specifically this is achieved by learning the cover of the geometric datasets. The performance of existing methods such as the 1D Mapper or Differentiable Mapper is sensitive to the choice to hyper parameters. To tackle this issue, the author proposed a non-convex optimization program with a new loss function that does not involve the filter function anymore. The new loss consists of three terms that try to ensure the resulting cover learned are satisfying from different aspects. Empirical results demonstrate the proposed method is able to output meaningful and competetive cover compared to SOTA.

**Claims And Evidence:**

Yes. All the claims are supported by either empirical result or theoretical analysis.

**Essential References Not Discussed:**

I don't aware there are any outstanding missing reference in the manuscript

**Experimental Designs Or Analyses:**

The current result looks to be sound. However, as most of the results are visualization of the output from different methods, there are several extra experiments (apologize if I missed something) that could be further investigated to enhance the soundness of the proposed algorithm:

1. As the author stated that initialization is important, could we provide more evidence on this, e.g. what are the results with/without a good initialization, or with different clustering methods for initialization
2. Similarly, can we have some plot that compares the embedding g at initialization and after the optimization? In experiments I feel there are some cases (e.g. MNIST) where spectral clustering is already able to provide a very meaningful result.
3. As in table 4, looks like the topology loss contributes most to the runtime, could we have some comparison on the output with different weights on the loss function components, which can further help us understand the significance of these proposed components.
4. Besides visualizing the output, is there anyway to perform some downstream tasks, e.g. clustering based on the optimized embedding (representation) to further showcase the strengths of the proposed methods.

**Methods And Evaluation Criteria:**

I think the proposed method appears to be sound and effective, as supported by several experiments on different datasets.

**Other Comments Or Suggestions:**

N/A

**Other Strengths And Weaknesses:**

Overall, I think this paper is well-written and easy to follow. The proposed method appears to be novel and sound. My main concern is on the experiments which has been elaborated in previous sections.

**Questions For Authors:**

N/A

**Relation To Broader Scientific Literature:**

The key contribution of this paper is proposed a new topology representation learning method, which could benefit various downstream tasks such as clustering and synchronization.

**Theoretical Claims:**

I didn't check the correctness of the theoretical claims but they appear to have no issue.

---

> ### Author Rebuttal · Authors · 2025-03-27
>
> Thank you for your thoughtful feedback!
>
> - About most of the results being visualizations: Three out of five experiments concern visualizations.
> We just want to emphasize the fact that the two other experiments concern topological inference, with experiment A being a quite thorough comparison with the main competitors based on sparse simplicial complexes.
>
> - About evidence on initialization being important:
> This will be addressed in the ablation study and sensitivity analysis.
> Please see "Main comment A", below.
>
> - About comparing initialization and output after optimization:
> This will be addressed too.
> Please see "Main comment A" below.
> Please also see "Main comment B", below, about differences between the clustering used for initialization and our output cover.
>
> - About using different weights for the loss components:
> Please see "Main comment A", below
>
> - About downstream tasks:
> Please see "Main comment C", in the response to reviewer 5NTw.
>
> **Main comment A: Ablation study and sensitivity analysis**
>
> We will add the requested ablation study and sensitivity analysis.
> We now outline our preliminary findings, and please see below for a sample experiment supporting these.
> We also emphasize that, in the paper, our default parameters work well in a wide range of datasets.
> - Initialization (random vs clustering).
> In terms of quality of output, this does not play a big role for small datasets (eg 2-sphere), but it does for larger data (eg MNIST). In terms of convergence speed, clustering initialization always leads to faster convergence (order of x10 or more for large datasets).
> - Ablation.
>     - Measure loss and Regularization loss: These are required for obtaining reasonable results.
>     - Topological loss: When using random initialization, it is required. When using a good clustering as initialization, the topological loss does not play a big role, and good results can be obtained without it.
>     - Geometry loss:
>     In our experiments thus far, it does not play an important role.
>     We believe that regular geometry is already enforced by the regularization loss; this is explained in Appendix F.2.
> - Sensitivity to parameters.
>     - Number of cover elements: The algorithm is robust to this choice.
>     This parameter is just an upper bound, since the algorithm is allowed to return empty cover elements which are just discarded.
>     If one wants to force more cover elements to be non-empty, one can increase the weight of the measure loss (since measure loss enforces many small sets as opposed to few large ones).
>     - Regularization weight: The algorithm is robust to this choice.
>     A large value will enforce larger cover elements, which can be used to have larger intersections between cover elements, in case the output is too disconnected.
>     - Number of neighbors for nearest neighbor graph: The algorithm is robust to this choice, as is usually the case with this parameter in unsupervised learning algorithms.
>     - Threshold (lambda):
>     This parameter is only required for producing a single graph (or simplicial complex), for, eg visualization, and it is not fixed for topological inference.
>     The algorithm is sensitive to this parameter, and the number of edges (intersections between cover elements) can change significantly when going from lambda = 1 to lambda = 0.
>     The default 0.5 usually leads to good results, but sometimes there might be too many intersections for easy visualization; in that case the solution is to increase lambda (as in Fig. 4 and Fig. 5).
>
> Sample experiment (the experiments in the paper will be more thorough): We run our method on the 2- and 3-sphere datasets and quantify topology recovery as in Table 1
> - Initialization and topology loss:
>     - Results can be reproduced with clustering initialization and without topology loss.
>     - With random initialization and no topology loss, topological recovery always fails.
>     - With random initialization and topology loss, recovery is successful most of the times (>80%).
> - Geometry loss: The results in Table 1 can be reproduced without the geometry loss.
> - Number of cover elements: Topology recovery is still successful for much larger values than the ones used in the Table (>30).
> - Regularization weight: Results are replicated with smaller and larger values (5, 10, 20).
> - Number of neighbors: Results are replicated with smaller and larger values (8, 15, 30).
>
> **Main comment B: About the clustering used at initialization**
>
> Although initialization plays an important role for large datasets, it does not, by itself, solve the cover learning problem: There are no intersections between the clusters in a clustering, so the nerve is discrete and contains no topological information (above dim 0), meaning that a clustering by itself wouldn't be useful for topological inference (experiments A and B) or visualization of interesting geometric structure (eg flare structure in Fig. 5).

---

### Decision · Program_Chairs · 2025-05-01

**Decision:**

Accept (poster)

**Comment:**

This paper proposes an optimization method to learn a topological cover of to represent the large-scale topological structure of geometric datasets. It addresses the constraints of previous Topological Data Analysis (TDA) method such as Mapper graph, which explicitly requires a hand-picked filter function and other parameters. The proposed method learns a fuzzy cover through optimization, with theoretical guarantees, and outperforms standard TDA methods in both quantitative topological inference and qualitative topology visualization. Reviewers are satisfied with the rebuttal although their request for downstream applications was not addressed well.